

**Dynamics of soil aggregate-related stoichiometric characteristics with tea-planting age and**
**soil depth in the southern Guangxi of China**
Ling Mao, Shaoming Ye, Shengqiang Wang*
Forestry College, Guangxi University, Nanning, 530004, China
*Corresponding author.
E-mail address: shengqiang@gxu.edu.cn
**ABSTRACT**
Soil ecological stoichiometry offers a sort of effective way to explore the distribution, cycling,
limitation, and balance of chemical elements in tea plantation ecosystems. This study was aim to
explore how soil organic C (OC) and nutrient contents (total N (TN), total P (TP), $Ca^{2+}$, $Mg^{2+}$,
$Fe^{2+}$, and $Mn^{2+}$) as well as their stoichiometric ratios (C/N, C/P, N/P, Ca/Mg, and Fe/Mn) vary
with tea-planting age (8, 17, 25, and 43 years) and soil depth (0-10, 10-20, 20-40, and 40-60 cm)
at the aggregate scales in the southern Guangxi of China. Our results showed that tea-planting
age and soil depth significantly influenced soil stoichiometric characteristics in various sized
aggregates. In different aged tea plantations, soil OC, TN, and TP contents as well as C/N, C/P,
and N/P ratios significantly decreased as the soil depth increased. In addition, soil $Ca^{2+}$ and $Mg^{2+}$
contents were significantly lower in the surface soil layer than the deeper soil layer, whereas soil
$Fe^{2+}$ and $Mn^{2+}$ contents showed totally opposite trends, and no significant differences were
detected among different soil depths in Ca/Mg and Fe/Mn ratios. Tea-planting age could
influence the variations in soil stoichiometric characteristics, but such effects were more obvious
at the 0-40 cm soil depth in contrast to the 40-60 cm soil depth. At the 0-40 cm soil depth,
continuous planting of tea was beneficial for the cumulation of soil OC, total N (TN), $Fe^{2+}$, and
$Mn^{2+}$, whereas soil $Ca^{2+}$ and $Mg^{2+}$ were susceptible to leaching losses over time. Compared with
other tea-planting regions in China, soil C/N ratio was higher in this tea-planting region, whereas
soil C/P and N/P ratios were much lower, indicating the lower contents of soil OC and TN,



especially the TN. Therefore, an appropriate increase in the amount of N fertilizer should be
applied in this tea-planting region. During the process of tea planting, the losses of soil $Ca^{2+}$ and
$Mg^{2+}$, especially the $Ca^{2+}$ (as indicated by the decrease in soil Ca/Mg ratio), could lead to the soil
acidification. Soil acidification could reduce $Fe^{2+}$ absorption and enhance $Mn^{2+}$ uptake by tea
plant (as indicated by the increase in soil Fe/Mn ratio), thereby causing the aggravation of $Fe^{2+}$
insufficiency and the emergence of $Mn^{2+}$ toxicity to tea plant. Overall, this study improved the
understanding of soil OC and nutrient dynamics in tea plantation ecosystems, and also provided
supplementary information for soil ecological stoichiometry in global terrestrial ecosystems.
**KEYWORDS**
Tea-planting age; Soil depth; Soil aggregate; Ecological stoichiometry
**1. Introduction**
Ecological stoichiometry offers a sort of valid approach to explore the distribution, cycling,
restriction, and balance of nutrients in terrestrial ecosystems (Yu et al., 2019), and plays a critical
role in recognizing the influence factors and drive mechanisms in ecological processes (Su et al.,
2019). On the one hand, carbon (C) is the most commonly seen element in plants (Prescott et al.,
2020), and nitrogen (N) and phosphorus (P) are critical control factors for the growth of plants
(Krouk and Kiba, 2020). The relationships amongst the three different elements are coupled
(Elser et al., 2003). Soil C/N, C/P, and N/P ratios represent not only the equilibrium features of
soil C, N, and P, but also the dynamics of fertility characteristics during the process of soil
genesis (Bai et al., 2020). On the other hand, calcium (Ca), magnesium (Mg), iron (Fe), and
manganese (Mn) are pivotal metallic nutritive elements for the development of plants (Liu et al.,
2021a). Soil total Ca, Mg, Fe, and Mn may exceed the demand of a single plant by more than a
thousand-fold and cannot sensitively reflect the needs of plants (Miner et al., 2018), but the
available fractions of these nutrients may be insufficient or redundant, resulting in the
deficiencies or abundances of plant nutrients (Otero et al., 2013). Thus, soil exchangeable Ca and



Mg as well as available Fe and Mn generate significant effects on the development of plants. Soil
Ca/Mg ratio reflects the relative effectiveness of these two ions and influences the buffering
capacity and acidification process in soil (Yin et al., 2016). Moreover, maintaining a proper soil
Fe/Mn ratio is pivotal for healthy soil because a lower ratio may indicate that plants have
encountered Fe depletion and Mn poisoning (Wang et al., 2017a).

Over the past decade, soil stoichiometric characteristics (mainly C-N-P, rather than Ca-Mg

or Fe-Mn) has been broadly studied across the world (Tian et al., 2010; Yang et al., 2013; Zhang
et al., 2016; Yue et al., 2017; Yu et al., 2018; Qiao et al., 2020). A wide agreement exists
amongst these studies that soil depth is vital for the regulation of soil stoichiometric
characteristics. Substantial studies have identified the decreasing trend of soil organic C (OC),
total N (TN), and total P (TP) contents as the soil depth increased (Yue et al., 2017; Yu et al.,
2018; Qiao et al., 2020), whereas conflicting vertical patterns were discovered for soil C/N, C/P,
and N/P ratios. For instance, decreasing trend of the C/P and N/P ratios was observed as the soil
depth increased in the data of the 2$^{nd}$ soil investigation in China (Tian et al., 2010). Nevertheless,
larger C/N ratio in the deeper soil layer, not the surface soil layer, was identified in a mollisol
plain in the northeast China (Zhang et al., 2016). Moreover, the C/N ratio displayed no
remarkable change throughout different soil depths in an investigation of alpine grassland on the
Qingzang Plateau (Yang et al., 2013). As shown above, inconsistent vertical patterns have been
reported for the C-N-P stoichiometric ratios in different soil ecosystems. Meanwhile, these
studies were mainly focused on the regional or global scales, rather than on the aggregate scales.

Soil aggregates constitute the fundamental parts of soil structure, and various sized

aggregates exert different abilities in the supply and reserve of soil OC and nutrients (Six et al.,
2004). Thus, to improve the comprehension about the structure and function of soil ecosystems,
more efforts should be made to observe the soil stoichiometric characteristics at the aggregate
scales (Xu et al., 2019; Cui et al., 2021). In recent period, lots of studies have reported the OC,
TN, and TP distribution in various sized aggregates, but these studies are ended with different



results. To be specific, some studies revealed the significant increases in the OC, TN, and TP
contents as the aggregate size decreased (Sarker et al., 2018; Piazza et al., 2020). Nevertheless,
some other studies drew the totally opposite trends (Lu et al., 2019; Liu et al., 2021b). These
show that the changes of soil OC, TN, and TP at the aggregate scales have received great
attention, whereas soil exchangeable alkali cations (i.e., $Ca^{2+}$ and $Mg^{2+}$) and available
micronutrients (i.e., $Fe^{2+}$ and $Mn^{2+}$) are rarely investigated.
In the past century, under the remarkable increase in population pressure, continuous tillage
and overmuch deforestation resulted in the dramatic decrease in soil fertility level in the southern
Guangxi of China (Jiang et al., 2018). For the purpose of tackling these challenges, the Chinese
government has rolled out the Grain for Green program in the hope of alleviating land
deterioration via converting farmlands to forest lands or grass lands (Zeng et al., 2020). Since the
initiation of such program, the south part of Guangxi has initiated the mode of transforming
farmlands into tea (*Camellia sinensis* L.) plantations as per the local geography and natural
resources (Zhang et al., 2017). Tea, as a pivotal cash crop, is commonly cultivated in the
developing nations, particularly in China, India, Kenya, and Sri Lanka. China is the first nation
to plant tea across the globe, with the tea-planting area reaching 3.17 million hectares in 2020
and presenting an elevating trend in the future (Chinese Tea Committee, 2020). Guangxi has the
subtropic monsoon climate and marks the key tea-planting region in China. According to the
statistics from Chinese Tea Committee (2020), more than 80% tea plantations of Guangxi are
situated at impoverished counties, and tea-planting industry turns to be the staple industry on
which poor counties depend to throw off poverty.
Our past studies indicated that the landuse shift from farmlands to tea plantations could
ameliorate soil fertility level (Zheng et al., 2011). Nevertheless, during the process of tea
planting, the variations in soil stoichiometric characteristics are still unclear. Meanwhile, since
tea plant serves as a deep root plant, it is vital to reveal how stoichiometric characteristics change
with increasing soil depth in tea plantation ecosystems. Thus, the present study was carried out to



investigate how soil OC and nutrient contents as well as their stoichiometric ratios vary with
tea-planting age (8, 17, 25, and 43 years) and soil depth (0-10, 10-20, 20-40, and 40-60 cm) at
the aggregate scales (< 0.25, 0.25-1, 1-2, and > 2 mm). In addition, we assumed that the
responses of soil OC and nutrient contents and their stoichiometric ratios to tea-planting age
would be different amongst different soil depths.
**2. Materials and methods**
2.1. Experiment site
In January 2019, the present study was completed at the Hengxian Agriculture Experiment
Center of Guangxi University (altitude of 557-563 m and slope degree of 13-15 °) (Figure S1).
Subtropic monsoon climate is predominant. Yearly average rainfall and temperature register
1304 mm and 21.6 °C, separately. Exposed soil horizon occurs early in the Mesozoic, which
gradually formed the Ultisols agrotype (IUSS Working Group, 2014). As early as in 1960s, due
to the high economic value of tea (especially "*Baimao* tea"), massive hectares of farmlands were
developed to tea plantations in such region. In the tea-planting course, tillage method is no tillage
and tea-planting density is almost $6 \times 10^4$ plants ha$^{-1}$. Yearly fertilization regime has been
displayed in our past studies (Wang and Ye, 2020). In all tea plantations, herbicides were not
applied and yellow sticky boards were used to prohibit pests, because the color may attract pests
and get them stuck on the boards. In addition, all tea plants were subjected to slight pruning in
September each year.
2.2. Experiment design
In general, examining the same location persistently has been considered a quite effective
approach in the monitoring of the variations in soil with time (Sparling et al., 2003).
Nevertheless, the challenges in long-period soil monitoring have made it urgent to develop
substitutional approaches to research the changes of soil over time, amongst which the most
common approach is the 'space-for-time' alternative (Zanella et al., 2018). To be specific,
disperse sites ('space') in diverse developmental phases are identified simultaneously to obtain a



chronological sequence of ages ('time'). In accordance with the 'space-for-time' alternative,
disperse spots of increasing ages display alike initial status and synchro-sampling at these
disperse spots equals the re-sampling at the same spot in different ages.

In this study, such approach was used to explore the variations in soil stoichiometric

characteristics in a chronological sequence of tea plantations. In general, certain underlying
mixture effects exist in the spatial variations of soil, hence the present study manages to mitigate
such effects via choosing tea plantations, which were cultured with the same tea variety
("*Baimao* tea") with different planting ages (8, 17, 25, and 43 years), and were located at the
same unit associated with geomorphological status. Every tea-planting age was duplicated in
quintuplicate, and afterwards generated 20 tea plantations. Separation amongst these tea
plantations was completed with distances of > 800 m between each other, hence decreasing the
space self-correlation and avoiding the pseudo-replication. For every tea plantation ($S \approx 1 \times 10^4$
$m^2$), a plot ($S = 20$ m $\times$ 20 m) was randomly established with distance of > 50 m away from the
tea plantation margin.
2.3. Litterfall and soil sampling

For every plot, the 5 litterfall specimens had been acquired from the surface of soil in the 5

randomly chosen subplots ($S = 1$ m $\times$ 1 m), and afterwards were integrated into a composite
litterfall specimen. An overall the 20 (4 tea-planting ages $\times$ 5 replicates) composite litterfall
specimens were desiccated at the 80 °C until steady weight. Then, the weights of these
desiccated litterfall specimens were measured, and the litterfall C (Nelson and Sommers, 1996)
and N (Bremner, 1996) contents were detected. Soil sampling was completed in the same sites of
the litterfall sampling. For every plot, the 5 soil specimens had been acquired by a spade from
every soil layer (i.e., 0-10, 10-20, 20-40, and 40-60 cm) in the 5 subplots, and afterwards were
integrated into a composite soil specimen. An overall the 80 (4 tea-planting ages $\times$ 4 soil layers $\times$
5 replicates) composite soil specimens were gently separated into naturally formed aggregates,
which were subjected to filtration by a 5 mm sifter to realize the removals of small stones, coarse



roots, and macrofauna. After that, soil specimens were used for the aggregate separation. For
every plot, moreover, extra 5 soil specimens were randomly chosen via cutting rings (V = 100
cm$^{-3}$, Ø = 50.46 mm, and depth = 50 mm) from every soil layer to assess the bulk density, clay
(< 0.002 mm), pH, OC, and nutrients of bulk soil.
2.4. Soil aggregate separation

As per the process of wet screening, 250 g of every composite soil specimen was subjected

to filtration via the 2, 1, and 0.25 mm sieves in a successive way (Kemper and Chepil, 1965). To
be specific, the composite soil specimens were soaked by the aqua destillata for 15 min, and
afterwards were oscillated in the vertical direction for 15 min at the 1 s$^{-1}$ oscillating rate and 5 cm
amplitude. Consequently, we obtained 4 various sized aggregates, covering microaggregates (<
0.25 mm), fine (0.25-1 mm), medium (1-2 mm), and coarse (> 2 mm) macroaggregates. All of
the aggregates were desiccated and weighted, and then aggregate-related OC and nutrients were
detected.
2.5. Soil property analyses

Prior to the analyses of soil physical-chemical properties, soil specimens were subjected to

atmospheric drying under indoor temperature condition. According to the cutting ring method
(Lu, 2000), soil specimens were oven-dried at 105 °C to the stable weight in order to measure the
bulk density. Soil clay was detected by the hydrometer (TM-85, Veichi, China) (Lu, 2000). Soil
pH was detected by the glassy electrode (MT-5000, Ehsy, China), with the ratio of soil : water
(mass : volume) as 1 : 2.5 (Lu, 2000). Soil OC and TN were identified via the acid dichromate
wet oxidation method (Nelson and Sommers, 1996) and the micro-Kjeldahl method (Bremner,
1996), separately. Soil TP was identified via the molybdate blue colorimetry method (Bray and
Kurtz, 1945). Soil exchangeable alkali cations (i.e., $Ca^{2+}$ and $Mg^{2+}$) were abstracted by the
ammonium acetate ($CH_3COONH_4$) (Thomas, 1982). In short, 2.5 g of every aggregate fraction
was weighted into Erlenmeyer flask to blend with 50 mL 1 M $CH_3COONH_4$ (pH = 7.0). The
extract liquid was agitated for 30 min under 150 rpm, and afterwards subjected to filtration via




Whatman No. 2 V filtration paper (quantitative and ashfree). Soil available micronutrients (i.e.,
$Fe^{2+}$ and $Mn^{2+}$) were abstracted by the diethylenetriamine pentaacetic acid (DTPA) (Lindsay and
Norvell, 1978). In short, 10 g of every aggregate fraction was weighted into Erlenmeyer flask to
blend with 20 mL 0.005 M DTPA + 0.01 M $CaCl_2$ + 0.1 M TEA (triethanolamine) (pH = 7.0).
The extract liquid was agitated for 2 h under 180 rpm, and afterwards subjected to filtration.
Entire extractable metallic cations were detected by the atomic absorption spectrometer (AAS,
Shimadzu, Japan). In this study, 5 standard specimens (GBW-07401), 5 blank specimens, and 80
parallel specimens (accounted for 20% of the total soil specimens) were used to control quality,
and the error is controlled in 5%.
2.6. Calculations and statistics
The mean weight diameter (MWD, mm) was utilized to indicate the stability of soil
aggregates (Kemper and Chepil, 1965):
$$\mathrm{MWD} = \sum_{i=1}^{4} (\mathrm{X}_i \times \mathrm{M}_i),$$
in the formula, $X_i$ indicates the $i^{\mathrm{th}}$ size aggregates' mean diameter (mm) and $M_i$ indicates the
$i^{\mathrm{th}}$ size aggregates' proportion (% in weight).
In the present study, since tea-planting age and soil depth serve as the two main factors,
statistic analysis was conducted separately by aggregate size. SPSS 22.0 was used for statistic
analysis (Table S1). One-way analysis of variance (ANOVA) was taken for exploring the effect
of tea-planting age on the litterfall characteristics. Two-way ANOVA was taken for exploring
the effects of tea-planting age and soil depth on the soil characteristics. Besides that, Pearson
correlation analysis was utilized to test the relationships between pH and stoichiometric ratios
(i.e., Ca/Mg and Fe/Mn ratios) in bulk soil during the process of tea planting.
**3. Results**
3.1. Composition and stability of soil aggregates
At the 0-10 and 10-20 cm soil depths, continuous planting of tea resulted in remarkable
variations in the proportions of various sized aggregates, apart from the medium and fine



macroaggregates (Figure 1). To be specific, the proportions of coarse macroaggregates
remarkably rose within the first 17 years and afterwards remarkably dropped, whereas the
proportions of microaggregates displayed an opposite trend over time. Meanwhile, the greatest
value of soil MWD was identified in the tea plantations of 17 years (Figure 1). Notably, the role
of tea-planting age in the aggregate composition and stability is limiting at the 20-40 and 40-60
cm soil depths. Across the 4 tea-planting ages, the coarse macroaggregates were prevailing at the
0-10 cm soil depth, which accounted for 32.60%-53.18% of bulk soil. However, at the 10-20,
20-40, and 40-60 cm soil depths, the microaggregates were dominant, which accounted for
33.80%-49.51%, 42.12%-48.24%, and 44.80%-49.45%, respectively. These results showed that
the coarse macroaggregate proportions reduced while the microaggregate proportions elevated
with increasing soil depth.
3.2. Contents of soil C, N, and P

At the aggregate scales, soil OC (Figure 2) and TN (Figure 3) contents increased with

increasing aggregate size, but the distribution of soil TP (Figure 4) was even in various sized
aggregates. In the tea-planting course (8-43 years), the aggregate-related OC and TN contents at
the 0-10, 10-20, and 20-40 cm soil depths were significantly elevated by 22%-35% and
14%-24%, 11%-22% and 9%-17%, and 8%-18% and 9%-13%, respectively. Nevertheless, no
remarkable variation existed in the aggregate-related TP content. Furthermore, at the 40-60 cm
soil depth, the aggregate-related OC, TN, and TP contents did not show significant variations
over time. Regardless of the tea-planting age, decreasing trend of the aggregate-related OC, TN,
and TP contents was observed as the soil depth increased.
3.3. Stoichiometric ratios of soil C, N, and P

In this study, the increases in aggregate-related C/N (Table 2), C/P (Table 3), and N/P

(Table 4) ratios were accompanied by the increasing aggregate size. At the 0-10, 10-20, and
20-40 cm soil depths, aggregate-related C/N ratio did not show remarkable variation while
aggregate-related C/P and N/P ratios remarkably increased during the process of tea planting.



Moreover, there was little role of tea-planting age in the aggregate-related C/N, C/P, and N/P
ratios at the 40-60 cm soil depth. In different aged tea plantations, aggregate-related C/N, C/P,
and N/P ratios dropped as the soil depth increased. For example, at the 0-10 cm soil depth,
aggregate-related C/N, C/P, and N/P ratios across the 4 tea-planting ages fluctuated in
20.81-23.04, 28.81-37.07, and 1.31-1.67, respectively. Meanwhile, at the 40-60 cm soil depth,
aggregate-related C/N, C/P, and N/P ratios fluctuated in 16.41-20.74, 13.44-22.88, and
0.84-1.08, respectively.
3.4. Contents of soil alkali cations and micronutrients
At the aggregate scales, soil exchangeable alkali cations (i.e., $Ca^{2+}$ and $Mg^{2+}$) were mainly
distributed in the microaggregates (Figures 5 and 6). However, soil available micronutrients (i.e.,
$Fe^{2+}$ and $Mn^{2+}$) were mainly existed in the coarse macroaggregates (Figures 7 and 8). In the
tea-planting course (8-43 years), the aggregate-related $Ca^{2+}$ and $Mg^{2+}$ contents at the 0-10, 10-20,
and 20-40 cm soil depths were significantly reduced by 31%-38% and 10%-24%, 23%-27% and
9%-18%, and 10%-16% and 5%-8%, respectively. However, the aggregate-related $Fe^{2+}$ and
$Mn^{2+}$ contents were significantly elevated by 16%-27% and 6%-9%, 11%-15% and 4%-7%, and
7%-12% and 3%-5%, respectively. In addition, at the 40-60 cm soil depth, the contents of
aggregate-related exchangeable alkali cations and available micronutrients did not show
significant variations over time. Irrespective of the tea-planting age, increasing trend of the
aggregate-related $Ca^{2+}$ and $Mg^{2+}$ contents was observed with increasing soil depth, but the
aggregate-related $Fe^{2+}$ and $Mn^{2+}$ contents showed an opposite trend.
3.5. Stoichiometric ratios of soil alkali cations and micronutrients
In this study, soil Ca/Mg (Table 5) and Fe/Mn (Table 6) ratios were evenly distributed in
various sized aggregates. At the 0-10, 10-20, and 20-40 cm soil depths, aggregate-related Ca/Mg
ratio remarkably decreased while aggregate-related Fe/Mn ratio remarkably increased in the
tea-planting course. Moreover, there was little role of tea-planting age in the aggregate-related
Ca/Mg and Fe/Mn ratios at the 40-60 cm soil depth. In different aged tea plantations, no



variations were observed amongst different soil depths in aggregate-related Ca/Mg and Fe/Mn
ratios. For example, at the 0-10 cm soil depth, aggregate-related Ca/Mg and Fe/Mn ratios across
the 4 tea-planting ages ranged from 1.81 to 1.96 and 0.76 to 0.85, respectively. Meanwhile, at the
40-60 cm soil depth, aggregate-related Ca/Mg and Fe/Mn ratios ranged from 1.88 to 1.92 and
0.78 to 0.82, respectively.
**4. Discussion**
4.1. Composition and stability of soil aggregates
Tea-planting age significantly influenced the aggregate composition and stability at the 0-10
and 10-20 cm soil depths, whereas the effect at the 20-40 and 40-60 cm soil depths was quite
limited (Figure 1). In the early (8-17 years) period, tea planting was beneficial for the transition
from microaggregates to coarse macroaggregates at the 0-10 and 10-20 cm soil depths. By
comparison, in the middle (17-25 years) and late (25-43 years) periods, tea planting induced
coarse macroaggregate destruction and microaggregate release. According to the hierarchical
concept of soil aggregates (Six et al., 2004), the quality of plant litterfall returning to the soil
determines the distribution of decomposition products of litterfall in various sized aggregates,
which ultimately impacts the aggregate composition. In the early period of tea planting, tea
litterfall displayed greater availability (as indicated by the lower litterfall C/N ratio) (Table 1),
revealing that the decomposition products of litterfall were easily combined into the coarse
macroaggregates, hence fostering the formation of coarse macroaggregates (Tisdall and Oades,
1982). Reversely, in the middle and late periods of tea planting, tea plants naturally encountered
aging processes and litterfall was progressively subjected to humification, which induced the
decomposition of coarse macroaggregates into microaggregates (Six and Paustian, 2014).
Moreover, the reduced litterfall amount and covering area after 17 years of tea planting (Table 1)
enhanced the rainfall eluviation and artificial interferences (i.e., pruning of tea plants and
application of fertilizers), which also caused the destruction of coarse macroaggregates. In the
tea-planting course, variation in aggregate stability was indicated via the change of MWD value.



At the 0-10 and 10-20 cm soil depths, the MWD value was the greatest in the 17 years of tea
planting (Figure 1), which was associated with the highest proportions of coarse
macroaggregates in the 17-year tea plantations. These findings indicated that the 17-year tea
plantations exhibited stronger aggregate stability in contrast to other plantations at the 0-10 and
10-20 cm soil depths.

Regardless of the tea-planting age, coarse macroaggregates were dominant in the topsoil

(0-10 cm) while microaggregates were dominant in the subsoil (10-60 cm) (Figure 1), indicating
transformation of aggregate composition from coarse macroaggregate-prevailing to
microaggregate-prevailing with the increase in soil depth. Also, alike outcomes were
corroborated by Li et al. (2015) and Zhu et al. (2017) from studies on tea plantations in the
southwest Sichuan of China. In the present study, coarse macroaggregates were the prevailing
fractions in the topsoil, not the subsoil, which was attributed to the surface cumulation of soil OC
(Figure 2). As an essential cementing agent, soil OC could foster the formation of coarse
macroaggregates (Al-Kaisi et al., 2014). Moreover, the reduced proportions of coarse
macroaggregates as the soil depth increased were also because of the elevated soil compactness
(as indicated by the bulk density) (Table 1). Soil densification could prevent the growth of plant
roots, hence causing the activities of soil microorganisms decreased, especially soil fungi (Kurmi
et al., 2020). Reduced activities of soil fungi could diminish the production of polysaccharose
and glomalin-related soil protein (GRSP) from the fungal hyphae, hence inducing the
proportions of soil macroaggregates decreased (Ji et al., 2019). Likewise, as per our past studies
(Wang et al., 2017b; Zhu et al., 2019), soil microbial activities and GRSP content served as the
vital effects in the formation and stabilisation of soil macroaggregates, and presented the higher
levels in the topsoil compared with the subsoil in tea plantation ecosystems. With increasing soil
depth, the decrease in MWD value (Figure 1) was mainly related to the change of soil aggregate
composition, especially for the decomposition of coarse macroaggregates into microaggregates,
implying that the topsoil exhibited stronger aggregate stability in contrast to the subsoil.





4.2. Contents of soil C, N, and P

In this study, more contents of soil OC and TN could be detected in coarse macroaggregates

(Figures 2 and 3), which conformed to the findings of Six et al. (2004) that macroaggregates
were comprised of microaggregates via temporary binding agents (i.e., microbe- and
plant-originated polysaccharides, fungal mycelium, and plant roots); meanwhile,
macroaggregates could provide the protection for the organic matters (OMs), hence causing the
cumulation of OC and TN in macroaggregates. Unlike soil OC and TN, soil TP was evenly
distributed in various sized aggregates (Figure 4). Moreover, Bhatnagar and Miller (1985) also
detected alike outcomes from soil specimens subjected to fresh poultry manure treatments, and
propelled the causal links influencing the TP distribution in soil aggregates. Specifically, (i)
introduced P was firstly adsorbed by clay particulates in soil and clay particulates were
discrepant in various sized aggregates, and (ii) introduced P had selective absorptive properties
for various sized aggregates. According to our findings, stochasticity seems to be one probable
mechanism that sheds light on the TP distribution in soil aggregates.

Continuous planting of tea could positively affect the cumulation of soil OC and TN, but

such positive effects were more obvious at the 0-40 cm soil depth in contrast to the 40-60 cm soil
depth (Figures 2 and 3). In this study, soil OC and TN contents exhibited a significant growing
trend over time, which was possibly associated with the following mechanisms. First, many
long-period tests had demonstrated the proactive roles of manure and chemical fertilizer
applications in soil OM cumulation (Tong et al., 2009; Zhou et al., 2013). Similarly, in the
tea-planting course, growing soil OC and TN contents were probably caused by the applications
of substantial swine manure every year (12 Mg ha$^{-1}$ year$^{-1}$) in this tea-planting region (Wang and
Ye, 2020). Second, plants serve as the prime OM sources in soil via root exudates and litterfall
remains (Franklin et al., 2020). In the tea-planting course, soil OC and TN cumulation probably
occurred as a result of the growing root systems and the increasing amounts of aboveground
litterfall attained from trimmed branches and leaves. Third, no tillage could provide physical



protection for the OMs combined with soil aggregates, and then further improve soil OC and TN
sequestration (Wulanningtyas et al., 2021). Notably, although the positive correlations of OC and
TN contents with clay content in soil have been reported, the present study revealed that
significant increases in the OC and TN contents were accompanied by no significant variation in
the clay content during the process of tea planting (Table 1). Similarly, Li et al. (2015) and Wang
et al. (2018) discovered as well that the changes of soil OC and TN contents were not influenced
by the clay content over time in tea plantation ecosystems, mainly because soil OC and TN
contents primarily depend on fertilization, tillage, root exudates, and litterfall remains, but soil
clay content is mainly controlled by its parent material (Rakhsh et al., 2020). Unlike soil OC and
TN, regardless of the soil depth, no remarkable difference existed in soil TP content amongst
different aged tea plantations (Figure 4), which implied the resistance of soil TP content to the
change of tea-planting age. Also, past studies verified that soil TP content was not related to the
tea-planting age (Wu et al., 2018; Yan et al., 2018), as soil P primarily derives from the
weathering release of soil minerals, instead of the short-period biology cycle (Cui et al., 2019).
In tea plantation ecosystems, the decreasing OC, TN, and TP contents with increasing soil
depth (Figures 2, 3, and 4) coincided with some past findings in other ecosystems, such as tropic
forests, bushlands, and grasslands (Stone and Plante, 2014; Yu et al., 2018; Qiao et al., 2020). In
the present study, the higher contents of OC, TN, and TP in the topsoil were associated with the
higher OM input, in which the soil OM contents in the topsoil were enriched by the input of
surface tea litterfall, root debris and exudates, and swine manure.
4.3. Stoichiometric ratios of soil C, N, and P
Soil C/N, C/P, and N/P ratios act as vital indicators of soil health (Liu et al., 2018), which
can be employed for exploring C circulation and guiding the equilibrium between N and P in soil
ecosystems (Sardans et al., 2012). In this study, soil C/N ratio grew with growing aggregate size
(Table 2), which indicated that the OMs in macroaggregates were younger and more unstable in
contrast to microaggregates (Six et al., 2004). Meanwhile, the OMs associated with



microaggregates experienced more degradation, resulting in the lower C/N ratio in the
microaggregates (Xu et al., 2019). In different aged tea plantations, soil OC and TN were
predominantly distributed in the coarse macroaggregates (Figures 2 and 3), but the TP was
evenly distributed in various sized aggregates (Figure 4). As a result, the associations of C/P and
N/P ratios to aggregate size primarily depended on the relationships of OC and TN contents with
aggregate size (Tables 3 and 4). As far as we know, the changes of soil C/P and N/P ratios at the
aggregate scales are rarely examined, although these kinds of knowledge are imperative because
of the biogeochemical cycles of N and P being influenced by the dynamics of soil aggregates
(Cui et al., 2021). Consequently, the impact generated by the aggregate size on the C/P and N/P
ratios ought to be studied more for the accurate forecast of soil N and P cycling under natural or
man-intervened ecosystems.

Irrespective of the soil depth, soil C/N ratio showed little significant variation in the

tea-planting course (Table 2). Meanwhile, tea-planting age significantly affected soil C/P and
N/P ratios at the 0-40 cm soil depth, not the 40-60 cm soil depth (Tables 3 and 4). Soil C/N ratio
is generally treated as the critical indicator which affects the formation and degradation of soil
OMs (Khan et al., 2016). Since response of soil TN content to soil environment change is almost
the same as soil OC content (Wang et al., 2018), soil C/N ratio did not show significant
difference amongst different aged tea plantations (Table 2). Likewise, Zhou et al. (2018) proved
that no close correlation existed between soil C/N ratio and vegetation coverage, because C and
N are structure elements and their cumulation and consumption in soil remain relative
consistency. Soil C/P ratio is the indicator suggesting P effectiveness, and higher C/P ratio often
denotes lower P effectiveness (Khan et al., 2016). In acidic soil (Table 1), available P was
adsorbed on the surfaces of Fe/Al oxides and clay minerals in a preferential way, because Fe/Al
oxides and clay minerals with greater surface areas could afford enough sites to available P
adsorption (Wu et al., 2018). As the tea-planting age increased, therefore, soil acidification led to
the decrease in P effectiveness (evidenced by the significant increase in soil C/P ratio) (Table 3).



Soil N and P are the prohibiting factors mostly seen during the process of plant growth, and thus,
N/P ratio can be utilized as one efficient indicator that shows nutrient restriction (Khan et al.,
2016). In this study, soil N/P ratio remarkably increased in the tea-planting course (Table 4),
mainly because soil TN content experienced significant increase while no such significant
change was found in TP content over time.
Soil C/N ratio decreased with increasing soil depth, regardless of the tea-planting age
(Table 2), which coincided with the majority of studies (Cao et al., 2015; Feng and Bao, 2017,
Yu et al., 2019). Batjes (1996) suggested that the decrease in soil C/N ratio as the soil depth
increased was triggered by the stratification of humic substance in the soil profile. Moreover, in
this study, the lower soil C/P and N/P ratios in the subsoil (Tables 3 and 4) backed the outcomes
of past studies in terrestrial ecosystems of China, which were on the foundation of the data from
both the 2nd soil investigation in China (Tian et al., 2010) and the Chinese Ecosystem Research
Network (CERN) (Chai et al., 2015).
Across the 4 tea-planting ages, the mean contents of OC and TN in bulk soil (0-20 cm) were
16.70 and 0.77 g kg$^{-1}$, separately, which were below the mean contents of OC (21.30 g kg$^{-1}$) and
TN (2.17 g kg$^{-1}$) in Chinese tea plantations (Sun et al., 2020; Xie et al., 2020). Moreover, in this
tea-planting region, the mean content of TP in bulk soil (0-20 cm) was 0.57 g kg$^{-1}$, corresponding
to the moderate level in Chinese tea plantations, where TP content varied in the range of
0.35-1.20 g kg$^{-1}$ (Wu et al., 2018; Sun et al., 2020). Herein, soil C/N ratio is higher compared
with other tea-planting regions in China, whereas soil C/P and N/P ratios are much lower (Sun et
al., 2020). These findings are primarily associated with the lower contents of soil OC and TN,
especially the TN. In general, N is the most limiting element in the net primary production of tea
plantation ecosystems (Miner et al., 2018), and this phenomenon also appeared in the southern
Guangxi of China.
4.4. Contents of soil alkali cations and micronutrients
According to the findings from Adesodun et al. (2007) and Emadi et al. (2009), the higher



contents of exchangeable alkali cations (including $Ca^{2+}$ and $Mg^{2+}$) were detected in both 2-4.76
and < 0.25 mm aggregates in the non-tillage soil. In the tillage course, however, the contents of
these two cations decreased in the 2-4.76 mm aggregates and increased in the < 0.25 mm
aggregates, revealing that the tillage practice could cause soil $Ca^{2+}$ and $Mg^{2+}$ to redistribute in
various sized aggregates. In comparison, the present study exhibited that the distribution of soil
$Ca^{2+}$ and $Mg^{2+}$ in aggregates was similar in different aged tea plantations (Figures 5 and 6),
implying that the distribution of these two cations in aggregates was seldom influenced by the
tea-planting age. To be specific, coarse macroaggregates had the lowest contents of $Ca^{2+}$ and
$Mg^{2+}$, whereas microaggregates exhibited the highest contents. These findings could be ascribed
to the larger specific surface areas of microaggregates (Adesodun et al., 2007), which increased
microaggregates' adsorption to $Ca^{2+}$ and $Mg^{2+}$ derived from root exudates, litterfall remains, and
manure (Emadi et al., 2009). Unlike exchangeable alkali cations, the contents of soil available
micronutrients (including $Fe^{2+}$ and $Mn^{2+}$) usually correspond to the contents of soil OMs (Wang
et al., 2017a), which are more abundant in macroaggregates (Six et al., 2004). Similarly, this
study also found that the $Fe^{2+}$ and $Mn^{2+}$ had a similar distribution pattern with OC at the
aggregate scales (Figures 7 and 8). Since the decomposition products of litterfall can be easily
integrated to the coarse macroaggregates (Six et al., 2004), the nutrient cycling of plant-soil
systems might lead to the higher contents of soil $Fe^{2+}$ and $Mn^{2+}$ in the coarse macroaggregates
(Wang et al., 2017a).

At the 0-40 cm soil depth, the contents of soil $Ca^{2+}$ and $Mg^{2+}$ significantly decreased over

time (Figures 5 and 6), which might be due to the applications of urea and $NH_4^+$-N fertilizer in
the tea-planting course for increasing tea leaf outputs. Urea hydrolysis can promote the
production of ammonium ions which are readily nitrified into nitrate, and the excessive proton
produced by the nitrification can compete for the adsorption sites with $Ca^{2+}$ and $Mg^{2+}$ (Wang et
al., 2017a). As a result, these cations were easy to lose from soil in the manner of leaching.
Except at the 40-60 cm soil depth, continuous planting of tea led to the remarkable increases in



soil $Fe^{2+}$ and $Mn^{2+}$ contents, which were elevated by 7%-27% and 3%-9% from 8 to 43 years of
tea planting, separately (Figures 7 and 8). This phenomenon was possibly caused by the soil
acidification (Table 1), which stimulates the release of soil $Fe^{2+}$ and $Mn^{2+}$ by mineralization and
desorption from soil OMs and minerals (Wang et al., 2017a). Tea, as an aluminium (Al)
cumulating crop, is able to cumulate Al in leaves (Li et al., 2016). Soil acidification in the
tea-planting course was due to the substantial tea litterfall into the soil annually via trimmed
branches and leaves (Li et al., 2016). At the same time, the rhizosphere deposition of massive
organic acids (i.e., malate, lemon acid, and oxalate acid) around the tea roots could provoke
localized acidification (Xue et al., 2006). In addition, for increasing the output of tea, tea
plantations needed to apply N fertilizers (i.e., urea and $NH_4^+$-N), thus leading to soil acidification
by the $NH_4^+$ nitration (Yang et al., 2018).

Across the 4 tea-planting ages, the contents of soil $Fe^{2+}$ and $Mn^{2+}$ were higher in the topsoil

than the subsoil (Figures 7 and 8), primarily owing to the usage of swine manure and the inputs
of tea litterfall and roots in the topsoil (Miner et al., 2018). Nevertheless, the contents of soil
$Ca^{2+}$ and $Mg^{2+}$ showed an opposite trend as the soil depth increased (Figures 5 and 6), because
soil $Ca^{2+}$ and $Mg^{2+}$ were easy to move from topsoil to subsoil in the manner of leaching (Hansen
et al., 2017).
4.5. Stoichiometric ratios of soil alkali cations and micronutrients

Tea-planting age exerted a remarkable influence on the Ca/Mg and Fe/Mn ratios at the 0-40

cm soil depth, not the 40-60 cm soil depth (Tables 5 and 6). To be specific, a notable decline in
the Ca/Mg ratio was found at the 0-40 cm soil depth over time (Table 5). From 8 to 43 years of
tea planting, the contents of $Ca^{2+}$ and $Mg^{2+}$ at the 0-40 cm soil depth decreased by 10%-38% and
5%-24%, separately (Figures 5 and 6), which revealed that the role of tea-planting age in the
content of soil $Ca^{2+}$ was greater than that of soil $Mg^{2+}$. Lu et al. (2014) suggested that the
selective losses of soil exchangeable alkali cations ($Ca^{2+}$ > $Mg^{2+}$) could lead to the
disequilibrium of soil metal ions in forest ecosystems. Similarly, in this study, the preferential



loss of soil $Ca^{2+}$ relative to $Mg^{2+}$ was the prime cause of the notable decline in the soil Ca/Mg
ratio in the tea-planting course. The depletion of soil exchangeable alkali cations (especially the
$Ca^{2+}$) could lead to the decrease in soil buffering capacity and soil acidification (Hansen et al.,
2017). Thus, the Ca/Mg ratio at the 0-40 cm soil depth was positively related ($P < 0.05$) to soil
pH across the 4 tea-planting ages (Figure 9). Soil acidification accelerated the mineralization and
desorption of soil available micronutrients from soil OMs and minerals (Wang et al., 2017a),
conducive to the significant increases in $Fe^{2+}$ and $Mn^{2+}$ contents at the 0-40 cm soil depth,
especially the $Fe^{2+}$ (Figures 7 and 8). In a chronological sequence of tea plantations, the negative
relationship ($P < 0.05$) of soil Fe/Mn ratio with soil pH in different soil depths (Figure 9)
indicated more cumulation of soil $Fe^{2+}$ relative to $Mn^{2+}$ over time (Table 6). Moreover, the
change of soil Fe/Mn ratio was also triggered by the antagonistic relationship between soil $Fe^{2+}$
and $Mn^{2+}$ during the process of tea plant uptake (Wang et al., 2017a). Tian et al. (2016)
discovered that soil acidification could reduce $Fe^{2+}$ absorption and enhance $Mn^{2+}$ uptake by
various plant species, thereby causing the increase in soil Fe/Mn ratio and threatening plant
productivity.
**5. Conclusions**
Herein, soil OC, TN, and TP contents as well as C/N, C/P, and N/P ratios decreased as the
soil depth increased. Moreover, soil $Ca^{2+}$ and $Mg^{2+}$ contents were lower in the topsoil than the
subsoil, whereas soil $Fe^{2+}$ and $Mn^{2+}$ contents showed an opposite trend, and no differences were
detected amongst different soil depths in soil Ca/Mg and Fe/Mn ratios. Tea-planting age could
influence the variations in soil OC and nutrient contents and their stoichiometric ratios, but such
effects were more obvious at the 0-40 cm soil depth in contrast to the 40-60 cm soil depth, thus
supporting our hypothesis. At the 0-40 cm soil depth, continuous planting of tea was favorable to
the cumulation of soil OC, TN, $Fe^{2+}$, and $Mn^{2+}$, whereas soil $Ca^{2+}$ and $Mg^{2+}$ were susceptible to
leaching losses over time. Compared with other tea-planting regions in China, soil C/N ratio is
higher in this tea-planting region, whereas soil C/P and N/P ratios are much lower, indicating



that soil OC and TN contents in the present study were lower, especially the TN. Therefore, an
appropriate increase in the amount of N fertilizer should be applied in this tea-planting region. In
the tea-planting course, the losses of soil $Ca^{2+}$ and $Mg^{2+}$, especially the $Ca^{2+}$ (as indicated by the
decrease in soil Ca/Mg ratio), could lead to the soil acidification. Soil acidification could reduce
$Fe^{2+}$ absorption and enhance $Mn^{2+}$ uptake by tea plant (as indicated by the increase in soil Fe/Mn
ratio), thereby causing the aggravation of $Fe^{2+}$ insufficiency and the emergence of $Mn^{2+}$ toxicity
to tea plant. Overall, the present study improved the understanding of soil OC and nutrient
dynamics in tea plantation ecosystems, and also provided supplementary information for soil
ecological stoichiometry in global terrestrial ecosystems.
**Data availability**
The data supporting the discovered information here can be presented by the relevant author
based on reasonable requests.
**Author contribution**
S.W. and S.Y. designed the experiments; L.M. carried out the experiments; S.W. and L.M.
analyzed the experimental results; L.M., S.W. and S.Y. wrote and edited the manuscript.
**Competing interests**
The authors declare no conflict of interest.
**Acknowledgments**
This study was funded by the National Natural Science Foundation of China (No.

31460196).

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





**Table 1** Physical-chemical properties of litterfall and soil in different aged tea plantations.

| Item | Soil depth | Tea-planting age | | | |
|---|---|---|---|---|---|
| | | 8 years | 17 years | 25 years | 43 years |
| Litterfall amount (g m$^{-2}$) | | 821 ± 21 B | 974 ± 34 A | 786 ± 28 C | 648 ± 19 D |
| Litterfall C/N ratio | | 14.23 ± 1.61 C | 12.68 ± 1.26 C | 17.32 ± 2.24 B | 21.37 ± 3.11 A |
| Soil bulk density (g cm$^{-3}$) | 0-10 cm | 1.28 ± 0.02 Ab | 1.20 ± 0.02 Bc | 1.26 ± 0.01 Ad | 1.31 ± 0.04 Ab |
| | 10-20 cm | 1.30 ± 0.03 Aab | 1.22 ± 0.03 Bc | 1.30 ± 0.03 Ac | 1.29 ± 0.02 Ab |
| | 20-40 cm | 1.32 ± 0.04 Aab | 1.31 ± 0.01 Ab | 1.34 ± 0.01 Ab | 1.33 ± 0.04 Ab |
| | 40-60 cm | 1.36 ± 0.01 Aa | 1.37 ± 0.02 Aa | 1.39 ± 0.02 Aa | 1.38 ± 0.03 Aa |
| Soil clay (%) | 0-10 cm | 34.69 ± 3.21 Aa | 35.91 ± 2.77 Aa | 33.12 ± 2.46 Aa | 35.08 ± 2.41 Aa |
| | 10-20 cm | 34.88 ± 2.08 Aa | 32.59 ± 3.02 Aa | 34.92 ± 3.67 Aa | 32.35 ± 2.68 Aa |
| | 20-40 cm | 35.26 ± 1.45 Aa | 34.57 ± 4.12 Aa | 34.51 ± 3.21 Aa | 34.29 ± 3.54 Aa |
| | 40-60 cm | 34.78 ± 3.66 Aa | 36.89 ± 2.98 Aa | 33.68 ± 1.91 Aa | 35.81 ± 3.69 Aa |
| Soil pH | 0-10 cm | 4.57 ± 0.02 Aa | 4.49 ± 0.01 Ba | 4.31 ± 0.03 Ca | 4.15 ± 0.02 Da |
| | 10-20 cm | 4.55 ± 0.03 Aa | 4.50 ± 0.01 Aa | 4.33 ± 0.02 Ba | 4.17 ± 0.02 Ca |
| | 20-40 cm | 4.60 ± 0.04 Aa | 4.53 ± 0.02 Ba | 4.34 ± 0.04 Ca | 4.19 ± 0.03 Da |
| | 40-60 cm | 4.58 ± 0.02 Aa | 4.54 ± 0.03 Aa | 4.32 ± 0.01 Ba | 4.21 ± 0.01 Ca |

Data represent the average of 5 replicates ± standard deviations. Different capital letters indicate significant
differences ($P < 0.05$) among different tea-planting ages. Different lower case letters indicate significant differences
($P < 0.05$) among different soil depths.





**Table 2** Effects of tea-planting age and soil depth on the soil C/N ratio in various sized aggregates.

| Sample | Soil depth | Tea-planting age | | | |
|---|---|---|---|---|---|
| | | 8 years | 17 years | 25 years | 43 years |
| Bulk soil | 0-10 cm | 22.21 ± 0.12 Aa | 21.72 ± 0.11 Aa | 22.06 ± 0.06 Aa | 21.99 ± 0.07 Aa |
| | 10-20 cm | 21.73 ± 0.08 Aa | 21.98 ± 0.08 Aa | 21.10 ± 0.13 Aa | 21.47 ± 0.11 Aa |
| | 20-40 cm | 18.86 ± 0.13 ABb | 18.07 ± 0.09 ABb | 17.52 ± 0.06 Bc | 19.38 ± 0.08 Ab |
| | 40-60 cm | 18.48 ± 0.04 Ab | 18.33 ± 0.09 Ab | 19.40 ± 0.04 Ab | 18.92 ± 0.05 Ab |
| | Mean | 20.32 | 20.03 | 20.02 | 20.44 |
| > 2 mm aggregates | 0-10 cm | 21.11 ± 0.13 Bab | 22.19 ± 0.08 ABa | 23.04 ± 0.10 Aa | 22.24 ± 0.11 ABa |
| | 10-20 cm | 22.39 ± 0.04 Aa | 22.97 ± 0.11 Aa | 21.92 ± 0.07 Ab | 21.83 ± 0.03 Aab |
| | 20-40 cm | 19.71 ± 0.12 Ab | 20.21 ± 0.04 Ab | 19.51 ± 0.14 Ac | 20.40 ± 0.04 Ab |
| | 40-60 cm | 20.74 ± 0.07 Aab | 19.61 ± 0.13 Ab | 19.67 ± 0.08 Ac | 20.51 ± 0.09 Ab |
| | Mean | 20.99 | 21.24 | 21.03 | 21.24 |
| 1-2 mm aggregates | 0-10 cm | 22.53 ± 0.07 Aa | 21.00 ± 0.12 Aa | 21.31 ± 0.08 Aa | 22.51 ± 0.07 Aa |
| | 10-20 cm | 22.09 ± 0.04 Aa | 22.72 ± 0.09 Aa | 22.69 ± 0.07 Aa | 22.55 ± 0.08 Aa |
| | 20-40 cm | 19.05 ± 0.05 Ab | 18.48 ± 0.11 Ab | 19.29 ± 0.04 Ab | 18.71 ± 0.10 Ab |
| | 40-60 cm | 17.47 ± 0.06 Ac | 17.69 ± 0.10 Ab | 16.94 ± 0.05 Ac | 17.27 ± 0.06 Ab |
| | Mean | 20.29 | 19.97 | 20.06 | 20.26 |
| 0.25-1 mm aggregates | 0-10 cm | 22.05 ± 0.08 Aa | 21.70 ± 0.11 Aa | 21.60 ± 0.09 Aa | 21.61 ± 0.04 Aa |
| | 10-20 cm | 21.63 ± 0.09 Aa | 22.56 ± 0.07 Aa | 21.90 ± 0.04 Aa | 21.89 ± 0.10 Aa |
| | 20-40 cm | 18.95 ± 0.12 Ab | 18.27 ± 0.04 Ab | 19.01 ± 0.07 Ab | 18.03 ± 0.12 Ab |
| | 40-60 cm | 17.56 ± 0.13 Ab | 17.08 ± 0.06 Ab | 17.48 ± 0.06 Ac | 17.22 ± 0.05 Ab |
| | Mean | 20.05 | 19.90 | 20.00 | 19.69 |
| < 0.25 mm aggregates | 0-10 cm | 21.89 ± 0.08 Aa | 21.30 ± 0.07 Aa | 20.81 ± 0.11 Aa | 21.93 ± 0.09 Aa |
| | 10-20 cm | 21.51 ± 0.08 ABa | 21.12 ± 0.03 ABa | 20.27 ± 0.07 Ba | 22.83 ± 0.06 Aa |
| | 20-40 cm | 17.78 ± 0.02 Ab | 18.45 ± 0.13 Ab | 17.46 ± 0.06 Ab | 17.56 ± 0.04 Ab |
| | 40-60 cm | 16.52 ± 0.05 Ab | 16.92 ± 0.07 Ac | 17.03 ± 0.10 Ab | 16.41 ± 0.08 Ab |
| | Mean | 19.42 | 19.45 | 18.89 | 19.68 |

Data represent the average of 5 replicates ± standard deviations. Different capital letters indicate significant
differences ($P < 0.05$) among different tea-planting ages. Different lower case letters indicate significant differences
($P < 0.05$) among different soil depths.





**Table 3** Effects of tea-planting age and soil depth on the soil C/P ratio in various sized aggregates.

| Sample | Soil depth | Tea-planting age | | | |
|---|---|---|---|---|---|
| | | 8 years | 17 years | 25 years | 43 years |
| Bulk soil | 0-10 cm | 30.93 ± 1.02 Ba | 31.63 ± 1.45 Ba | 35.94 ± 1.41 Aa | 34.50 ± 2.89 Aa |
| | 10-20 cm | 23.60 ± 0.85 Bb | 24.18 ± 0.84 Bb | 26.28 ± 1.21 Ab | 26.56 ± 1.47 Ab |
| | 20-40 cm | 19.92 ± 0.48 Cc | 20.13 ± 0.71 BCc | 21.15 ± 0.89 Bc | 24.41 ± 0.98 Ab |
| | 40-60 cm | 17.41 ± 0.69 Ac | 17.21 ± 0.58 Ad | 18.12 ± 0.24 Ad | 17.59 ± 1.22 Ac |
| | Mean | 22.97 | 23.29 | 25.37 | 25.76 |
| > 2 mm aggregates | 0-10 cm | 32.04 ± 1.04 Ca | 32.14 ± 0.98 Ca | 35.54 ± 1.07 Ba | 37.07 ± 0.38 Aa |
| | 10-20 cm | 25.93 ± 0.84 Cb | 24.41 ± 1.07 Cb | 27.21 ± 0.37 Bb | 29.23 ± 0.98 Ab |
| | 20-40 cm | 22.13 ± 0.97 Bc | 21.43 ± 1.12 Bc | 26.33 ± 0.86 Ab | 27.29 ± 1.24 Ab |
| | 40-60 cm | 22.40 ± 2.02 Ac | 22.88 ± 0.87 Abc | 22.67 ± 1.24 Ac | 21.63 ± 1.56 Ac |
| | Mean | 25.62 | 25.21 | 27.94 | 28.81 |
| 1-2 mm aggregates | 0-10 cm | 29.52 ± 1.01 Da | 31.48 ± 0.47 Ca | 33.54 ± 0.97 Ba | 36.53 ± 0.81 Aa |
| | 10-20 cm | 24.76 ± 0.38 Bb | 26.58 ± 0.58 Ab | 27.55 ± 0.47 Ab | 26.68 ± 0.97 Ab |
| | 20-40 cm | 20.68 ± 1.14 Bc | 20.51 ± 1.48 Bc | 21.95 ± 1.05 Bc | 26.07 ± 0.78 Ab |
| | 40-60 cm | 16.78 ± 0.87 Bd | 18.04 ± 0.98 Ac | 16.63 ± 1.24 Bd | 17.55 ± 1.05 ABc |
| | Mean | 22.93 | 24.15 | 24.92 | 26.71 |
| 0.25-1 mm aggregates | 0-10 cm | 31.44 ± 1.27 Aa | 30.46 ± 0.78 Aa | 30.91 ± 1.08 Aa | 30.62 ± 0.98 Aa |
| | 10-20 cm | 23.60 ± 0.27 Bb | 25.38 ± 0.38 ABb | 24.41 ± 1.14 ABb | 26.41 ± 0.57 Ab |
| | 20-40 cm | 18.96 ± 1.47 Cc | 21.62 ± 0.45 Bc | 19.78 ± 0.87 Cc | 25.60 ± 1.02 Ab |
| | 40-60 cm | 18.18 ± 0.87 Ac | 17.40 ± 0.38 Ad | 17.43 ± 0.91 Ac | 17.92 ± 1.34 Ac |
| | Mean | 23.05 | 23.71 | 23.13 | 25.14 |
| < 0.25 mm aggregates | 0-10 cm | 28.81 ± 1.01 Ba | 30.60 ± 1.07 Aa | 29.78 ± 0.87 ABa | 31.00 ± 0.38 Aa |
| | 10-20 cm | 19.39 ± 1.17 Cb | 21.86 ± 0.68 Bb | 22.36 ± 0.78 ABb | 23.19 ± 0.98 Ab |
| | 20-40 cm | 15.22 ± 0.87 Bc | 17.11 ± 1.14 Ac | 17.46 ± 0.94 Ac | 18.50 ± 0.75 Ac |
| | 40-60 cm | 13.73 ± 0.74 Ac | 14.50 ± 0.74 Ad | 13.44 ± 1.00 Ad | 14.02 ± 0.91 Ad |
| | Mean | 19.29 | 21.02 | 20.76 | 21.68 |

Data represent the average of 5 replicates ± standard deviations. Different capital letters indicate significant
differences ($P < 0.05$) among different tea-planting ages. Different lower case letters indicate significant differences
($P < 0.05$) among different soil depths.



**Table 4** Effects of tea-planting age and soil depth on the soil N/P ratio in various sized aggregates.

| Sample | Soil depth | Tea-planting age | | | |
|---|---|---|---|---|---|
| | | 8 years | 17 years | 25 years | 43 years |
| Bulk soil | 0-10 cm | 1.39 ± 0.04 Ba | 1.46 ± 0.02 Ba | 1.63 ± 0.04 Aa | 1.57 ± 0.02 ABa |
| | 10-20 cm | 1.09 ± 0.01 Bb | 1.10 ± 0.04 Bb | 1.25 ± 0.08 Ab | 1.24 ± 0.03 Ab |
| | 20-40 cm | 1.06 ± 0.02 Bb | 1.11 ± 0.07 ABb | 1.21 ± 0.02 Ab | 1.26 ± 0.02 Ab |
| | 40-60 cm | 0.94 ± 0.06 Ab | 0.99 ± 0.06 Ab | 0.93 ± 0.01 Ac | 0.98 ± 0.04 Ac |
| | Mean | 1.12 | 1.17 | 1.25 | 1.26 |
| > 2 mm aggregates | 0-10 cm | 1.52 ± 0.05 ABa | 1.45 ± 0.01 Ba | 1.54 ± 0.04 ABa | 1.67 ± 0.02 Aa |
| | 10-20 cm | 1.16 ± 0.02 ABb | 1.06 ± 0.02 Bc | 1.24 ± 0.03 Ab | 1.34 ± 0.01 Ab |
| | 20-40 cm | 1.11 ± 0.03 Bb | 1.24 ± 0.02 ABb | 1.25 ± 0.05 ABb | 1.38 ± 0.04 Ab |
| | 40-60 cm | 1.08 ± 0.04 Ab | 1.07 ± 0.03 Ac | 1.00 ± 0.04 Ac | 1.06 ± 0.03 Ac |
| | Mean | 1.22 | 1.20 | 1.26 | 1.36 |
| 1-2 mm aggregates | 0-10 cm | 1.31 ± 0.03 Ba | 1.50 ± 0.01 Aa | 1.57 ± 0.02 Aa | 1.62 ± 0.04 Aa |
| | 10-20 cm | 1.12 ± 0.04 Ab | 1.17 ± 0.02 Ab | 1.21 ± 0.03 Ab | 1.18 ± 0.05 Abc |
| | 20-40 cm | 0.98 ± 0.01 Bc | 1.05 ± 0.01 Bb | 1.14 ± 0.04 ABbc | 1.26 ± 0.02 Ab |
| | 40-60 cm | 0.96 ± 0.06 Ac | 1.02 ± 0.03 Ab | 0.98 ± 0.06 Ac | 1.03 ± 0.03 Ac |
| | Mean | 1.09 | 1.19 | 1.23 | 1.27 |
| 0.25-1 mm aggregates | 0-10 cm | 1.43 ± 0.02 Aa | 1.40 ± 0.04 Aa | 1.43 ± 0.02 Aa | 1.42 ± 0.01 Aa |
| | 10-20 cm | 1.09 ± 0.02 Bb | 1.13 ± 0.01 ABb | 1.11 ± 0.01 ABb | 1.21 ± 0.01 Ab |
| | 20-40 cm | 0.96 ± 0.06 Bb | 1.02 ± 0.02 Bb | 1.13 ± 0.04 ABb | 1.29 ± 0.03 Ab |
| | 40-60 cm | 0.98 ± 0.04 Ab | 1.02 ± 0.03 Ab | 0.94 ± 0.02 Ac | 1.04 ± 0.02 Ac |
| | Mean | 1.11 | 1.14 | 1.15 | 1.24 |
| < 0.25 mm aggregates | 0-10 cm | 1.32 ± 0.04 Ba | 1.44 ± 0.03 Aa | 1.43 ± 0.02 Aa | 1.41 ± 0.04 Aa |
| | 10-20 cm | 0.90 ± 0.01 Bb | 1.04 ± 0.02 ABb | 1.10 ± 0.01 Ab | 1.02 ± 0.03 ABb |
| | 20-40 cm | 0.91 ± 0.04 Bb | 0.93 ± 0.02 Bbc | 1.00 ± 0.03 Ab | 1.05 ± 0.02 Ab |
| | 40-60 cm | 0.88 ± 0.03 Ab | 0.86 ± 0.04 Ac | 0.84 ± 0.02 Ac | 0.85 ± 0.04 Ac |
| | Mean | 1.00 | 1.06 | 1.09 | 1.08 |

Data represent the average of 5 replicates ± standard deviations. Different capital letters indicate significant
differences ($P < 0.05$) among different tea-planting ages. Different lower case letters indicate significant differences
($P < 0.05$) among different soil depths.



**Table 5** Effects of tea-planting age and soil depth on the soil Ca/Mg ratio in various sized aggregates.

| Sample | Soil depth | Tea-planting age | | | |
|---|---|---|---|---|---|
| | | 8 years | 17 years | 25 years | 43 years |
| Bulk soil | 0-10 cm | 1.94 ± 0.12 Aa | 1.91 ± 0.05 Aa | 1.86 ± 0.12 ABa | 1.82 ± 0.07 Bab |
| | 10-20 cm | 1.93 ± 0.08 Aa | 1.87 ± 0.07 ABa | 1.87 ± 0.08 ABa | 1.84 ± 0.12 Bab |
| | 20-40 cm | 1.96 ± 0.14 Aa | 1.90 ± 0.14 Ba | 1.88 ± 0.04 Ba | 1.80 ± 0.14 Cb |
| | 40-60 cm | 1.91 ± 0.11 Aa | 1.88 ± 0.09 Aa | 1.90 ± 0.07 Aa | 1.89 ± 0.06 Aa |
| | Mean | 1.94 | 1.89 | 1.88 | 1.84 |
| > 2 mm aggregates | 0-10 cm | 1.96 ± 0.17 Aa | 1.89 ± 0.08 ABa | 1.88 ± 0.16 ABa | 1.83 ± 0.04 Bab |
| | 10-20 cm | 1.92 ± 0.14 Aa | 1.87 ± 0.18 Ba | 1.86 ± 0.06 Ba | 1.88 ± 0.07 Bab |
| | 20-40 cm | 1.95 ± 0.08 Aa | 1.88 ± 0.06 ABa | 1.89 ± 0.07 ABa | 1.81 ± 0.12 Bb |
| | 40-60 cm | 1.90 ± 0.11 Aa | 1.91 ± 0.09 Aa | 1.89 ± 0.11 Aa | 1.90 ± 0.13 Aa |
| | Mean | 1.93 | 1.89 | 1.88 | 1.86 |
| 1-2 mm aggregates | 0-10 cm | 1.94 ± 0.20 Aa | 1.86 ± 0.17 Ba | 1.85 ± 0.08 Ba | 1.84 ± 0.14 Ba |
| | 10-20 cm | 1.95 ± 0.15 Aa | 1.90 ± 0.16 Ba | 1.88 ± 0.08 Ba | 1.87 ± 0.10 Ba |
| | 20-40 cm | 1.92 ± 0.07 Aa | 1.84 ± 0.05 Ba | 1.86 ± 0.12 Ba | 1.85 ± 0.08 Ba |
| | 40-60 cm | 1.90 ± 0.06 Aa | 1.89 ± 0.06 Aa | 1.88 ± 0.03 Aa | 1.89 ± 0.09 Aa |
| | Mean | 1.93 | 1.87 | 1.87 | 1.86 |
| 0.25-1 mm aggregates | 0-10 cm | 1.95 ± 0.14 Aa | 1.88 ± 0.17 Ba | 1.87 ± 0.06 Ba | 1.81 ± 0.07 Cb |
| | 10-20 cm | 1.93 ± 0.11 Aa | 1.90 ± 0.08 ABa | 1.86 ± 0.07 ABa | 1.84 ± 0.04 Bb |
| | 20-40 cm | 1.94 ± 0.12 Aa | 1.86 ± 0.10 ABa | 1.87 ± 0.03 ABa | 1.82 ± 0.06 Bb |
| | 40-60 cm | 1.92 ± 0.07 Aa | 1.90 ± 0.06 Aa | 1.91 ± 0.05 Aa | 1.89 ± 0.09 Aa |
| | Mean | 1.94 | 1.89 | 1.88 | 1.84 |
| < 0.25 mm aggregates | 0-10 cm | 1.92 ± 0.06 Aa | 1.85 ± 0.04 Ba | 1.83 ± 0.08 Ba | 1.82 ± 0.12 Bb |
| | 10-20 cm | 1.94 ± 0.17 Aa | 1.86 ± 0.12 Ba | 1.87 ± 0.08 Ba | 1.85 ± 0.07 Bab |
| | 20-40 cm | 1.91 ± 0.08 Aa | 1.85 ± 0.07 Ba | 1.84 ± 0.07 Ba | 1.86 ± 0.04 Bab |
| | 40-60 cm | 1.92 ± 0.11 Aa | 1.90 ± 0.03 Aa | 1.89 ± 0.06 Aa | 1.91 ± 0.13 Aa |
| | Mean | 1.92 | 1.87 | 1.86 | 1.86 |

Data represent the average of 5 replicates ± standard deviations. Different capital letters indicate significant
differences ($P < 0.05$) among different tea-planting ages. Different lower case letters indicate significant differences
($P < 0.05$) among different soil depths.





**Table 6** Effects of tea-planting age and soil depth on the soil Fe/Mn ratio in various sized aggregates.

| Sample | Soil depth | Tea-planting age | | | |
| --- | --- | --- | --- | --- | --- |
| | | 8 years | 17 years | 25 years | 43 years |
| Bulk soil | 0-10 cm | 0.78 ± 0.04 Ba | 0.82 ± 0.01 ABa | 0.81 ± 0.04 ABa | 0.84 ± 0.02 Aab |
| | 10-20 cm | 0.76 ± 0.02 Ba | 0.81 ± 0.04 Aa | 0.82 ± 0.01 Aa | 0.83 ± 0.04 Aab |
| | 20-40 cm | 0.75 ± 0.03 Ca | 0.80 ± 0.03 Ba | 0.81 ± 0.05 Ba | 0.85 ± 0.02 Aa |
| | 40-60 cm | 0.78 ± 0.05 Aa | 0.80 ± 0.05 Aa | 0.79 ± 0.03 Aa | 0.81 ± 0.03 Ab |
| | Mean | 0.77 | 0.81 | 0.81 | 0.83 |
| > 2 mm | 0-10 cm | 0.77 ± 0.02 Ba | 0.81 ± 0.01 ABa | 0.81 ± 0.03 ABa | 0.83 ± 0.02 Aab |
| aggregates | 10-20 cm | 0.75 ± 0.04 Ca | 0.80 ± 0.04 Ba | 0.79 ± 0.01 Ba | 0.84 ± 0.02 Aa |
| | 20-40 cm | 0.77 ± 0.01 Ba | 0.78 ± 0.01 Ba | 0.82 ± 0.02 Aa | 0.82 ± 0.03 Aab |
| | 40-60 cm | 0.78 ± 0.03 Aa | 0.81 ± 0.02 Aa | 0.79 ± 0.04 Aa | 0.80 ± 0.01 Ab |
| | Mean | 0.77 | 0.80 | 0.80 | 0.82 |
| 1-2 mm | 0-10 cm | 0.76 ± 0.01 Cab | 0.82 ± 0.02 ABa | 0.80 ± 0.01 Ba | 0.85 ± 0.03 Aa |
| aggregates | 10-20 cm | 0.75 ± 0.02 Bb | 0.81 ± 0.01 Aa | 0.82 ± 0.02 Aa | 0.84 ± 0.01 Aa |
| | 20-40 cm | 0.78 ± 0.02 Bab | 0.80 ± 0.03 ABa | 0.82 ± 0.01 ABa | 0.85 ± 0.04 Aa |
| | 40-60 cm | 0.79 ± 0.01 Aa | 0.82 ± 0.01 Aa | 0.81 ± 0.03 Aa | 0.82 ± 0.02 Aa |
| | Mean | 0.77 | 0.81 | 0.81 | 0.84 |
| 0.25-1 mm | 0-10 cm | 0.78 ± 0.03 Bab | 0.82 ± 0.02 ABa | 0.82 ± 0.03 ABa | 0.84 ± 0.01 Aab |
| aggregates | 10-20 cm | 0.77 ± 0.01 Bab | 0.81 ± 0.01 ABab | 0.82 ± 0.02 ABa | 0.85 ± 0.02 Aa |
| | 20-40 cm | 0.75 ± 0.02 Bb | 0.78 ± 0.04 ABb | 0.81 ± 0.05 ABa | 0.84 ± 0.01 Aab |
| | 40-60 cm | 0.80 ± 0.04 Aa | 0.79 ± 0.02 Aab | 0.80 ± 0.02 Aa | 0.81 ± 0.03 Ab |
| | Mean | 0.78 | 0.80 | 0.81 | 0.84 |
| < 0.25 mm | 0-10 cm | 0.79 ± 0.02 Ba | 0.81 ± 0.03 Ba | 0.81 ± 0.03 Ba | 0.85 ± 0.01 Aa |
| aggregates | 10-20 cm | 0.77 ± 0.01 Ba | 0.82 ± 0.02 Aa | 0.80 ± 0.01 ABa | 0.83 ± 0.03 Aa |
| | 20-40 cm | 0.78 ± 0.03 Ba | 0.80 ± 0.02 ABa | 0.82 ± 0.02 ABa | 0.83 ± 0.04 Aa |
| | 40-60 cm | 0.80 ± 0.01 Aa | 0.81 ± 0.03 Aa | 0.80 ± 0.04 Aa | 0.82 ± 0.02 Aa |
| | Mean | 0.79 | 0.81 | 0.81 | 0.83 |

Data represent the average of 5 replicates ± standard deviations. Different capital letters indicate significant
differences ($P < 0.05$) among different tea-planting ages. Different lower case letters indicate significant differences
($P < 0.05$) among different soil depths.



**Figure 1** Effects of tea-planting age and soil depth on the composition and stability of soil aggregates. Data represent the average of 5 replicates and error bars represent the standard deviations. Different capital letters indicate significant differences ($P < 0.05$) among different tea-planting ages. Different lower case letters indicate significant differences ($P < 0.05$) among different soil depths.

**Figure 2** Effects of tea-planting age and soil depth on the soil organic C content in various sized aggregates. Data represent the average of 5 replicates and error bars represent the standard deviations. Different capital letters indicate significant differences ($P < 0.05$) among different tea-planting ages. Different lower case letters indicate significant differences ($P < 0.05$) among different soil depths.

**Figure 3** Effects of tea-planting age and soil depth on the soil total N content in various sized aggregates. Data represent the average of 5 replicates and error bars represent the standard deviations. Different capital letters indicate significant differences ($P < 0.05$) among different tea-planting ages. Different lower case letters indicate significant differences ($P < 0.05$) among different soil depths.

**Figure 4** Effects of tea-planting age and soil depth on the soil total P content in various sized aggregates. Data represent the average of 5 replicates and error bars represent the standard deviations. Different capital letters indicate significant differences ($P < 0.05$) among different tea-planting ages. Different lower case letters indicate significant differences ($P < 0.05$) among different soil depths.

**Figure 5** Effects of tea-planting age and soil depth on the soil exchangeable $Ca^{2+}$ content in various sized aggregates. Data represent the average of 5 replicates and error bars represent the standard deviations. Different capital letters indicate significant differences ($P < 0.05$) among different tea-planting ages. Different lower case letters indicate significant differences ($P < 0.05$) among different soil depths.

**Figure 6** Effects of tea-planting age and soil depth on the soil exchangeable $Mg^{2+}$ content in various sized aggregates. Data represent the average of 5 replicates and error bars represent the standard deviations. Different capital letters indicate significant differences ($P < 0.05$) among different tea-planting ages. Different lower case letters indicate significant differences ($P < 0.05$) among different soil depths.

**Figure 7** Effects of tea-planting age and soil depth on the soil available $Fe^{2+}$ content in various sized aggregates. Data represent the average of 5 replicates and error bars represent the standard deviations. Different capital letters indicate significant differences ($P < 0.05$) among different tea-planting ages. Different lower case letters indicate significant differences ($P < 0.05$) among different soil depths.

**Figure 8** Effects of tea-planting age and soil depth on the soil available $Mn^{2+}$ content in various sized aggregates. Data represent the average of 5 replicates and error bars represent the standard deviations. Different capital letters indicate significant differences ($P < 0.05$) among different tea-planting ages. Different lower case letters indicate significant differences ($P < 0.05$) among different soil depths.

**Figure 9** Relationships of soil Ca/Mg and Fe/Mn ratios with soil pH in different soil layers. ** indicates significant differences at $P < 0.01$.




**Figure 1**

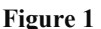

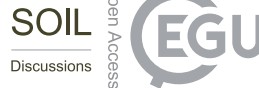

**Figure 2**

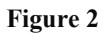



**Figure 3**

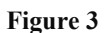



**Figure 4**

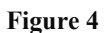

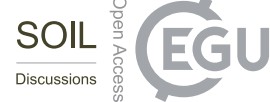



**Figure 5**





**Figure 6**

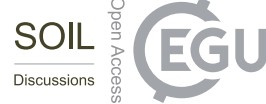



**Figure 7**

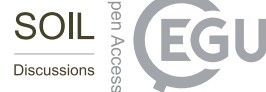

**Figure 8**







Figure 9

