# Peer review of "Soil nutrient stoichiometry varied with tea plantation age and soil depth at an aggregate"

_SOIL, 2021_

## Author Comment (AC1)

Dear editor Jocelyn Lavallee and reviewer #1,

We really appreciate you to give us the chance of revision. Thanks for your comments concerning our manuscript entitled "Dynamics of soil aggregate-related stoichiometric characteristics with tea-planting age and soil depth in the southern Guangxi of China" (SOIL-2021-147). We have made the corrections which we hope will meet with your approval. The revised portions are marked in blue ink in the paper. The main corrections and our responds to the comments are as follows.

**Reviewer #1:**

The manuscript (Ref. No. soil-2021-147) reported an interesting work on varied stoichiometric characteristics resulted from different tea growing age and soil depth. Such topic fits the scope of the journal very well.

Response: Thank you so much for your time and comments.

However, there are some concerns deserve further clarification before publication. The title needs to be polished due to the unclear expression, e.g., Stoichiometric characteristics of … varied with tea-planting age and soil depth at an aggregate scale in the southern Guangxi of China.

Response: Revised (L 1-2).

Numerous syntaxes and/or grammar problems or misuses existed in the current version, which makes great difficulties in understanding the main points. Native English editing service for the draft was strongly recommended.

Response: We have invited a native English speaker to edit the manuscript in order to improve the logical flow and make the relevant expressions more clear, and also carefully inspected and corrected the details such as word spelling, document information, and English grammar. Please see the revised manuscript.

The research needs or gap for the present study should be clearly indicated and justified as well as the work at the aggregate scale that maybe a potentially important innovative aspect.

Response: Revised (L 79-90). As the basic unites of soil structure, soil aggregates are complex ensembles composed of primary particles as well as organic matter (OM). According to the differences of binding agents, soil aggregates can be classified into microaggregates (< 0.25 mm) and macroaggregates (> 0.25 mm). In general, persistent binding agents (like humified OM and polyvalent metal cation complexes) contribute to the binding of primary particles into microaggregates. Differently, temporary binding agents (like fungal hyphae, plant roots, and polysaccharides) aggregating with microaggregates conduces to the formation of macroaggregates. As shown above, soil aggregates with various sizes exert different abilities in the supply and reserve of soil OC and nutrients. Thus, to improve the comprehension about the structure and function of

soil ecosystems, more efforts should be made to observe the soil stoichiometric characteristics at the aggregate scales.

Some detailed comments for your reference:

P1 Line9, "… a sort of effective way …" should be "… an effective way…".

Response: Revised (L 9).

P1 Line10, "this study was aim to…" changes to "the aim of this study was to…", or "this study was aimed to…", better?

Response: Revised (L 10).

P1 Line 15-16, in various sized aggregates should be in different sizes of aggregates. In different aged tea plantations? Confusing expression. Among different ages of tea gardens or cultivations?

Response: Revised (L 15-16).

P2 Line 27, "an appropriate increase" could be more quantitative or specific?

Response: We deleted this inaccurate sentence.

P2 Line 28, During the process of tea planting, tea growth, better?

Response: Revised (L 24).

P2 Line 31, tea plant should be tea plants or trees. Same as the remaining context.

Response: Revised (L 27 and elsewhere).

We believe that we have revised and improved this manuscript to the best of our abilities. In addition, we have made further changes according to the useful and helpful comments you have provided. We sincerely appreciate your time and effort on our behalf, and we truly hope that these corrections will meet with your approval.

Best regards,

Shengqiang Wang

---

## Author Comment (AC2)

Dear editor Jocelyn Lavallee and reviewers #2,

We really appreciate you to give us the chance of revision. Thanks for your comments concerning our manuscript entitled "Dynamics of soil aggregate-related stoichiometric characteristics with tea-planting age and soil depth in the southern Guangxi of China" (SOIL-2021-147). We have made the corrections which we hope will meet with your approval. The revised portions are marked in blue ink in the paper. The main corrections and our responds to the comments are as follows.

**Reviewer #2:**

The manuscript describes a study of soil chemistry within different soil aggregate sizes at various soil depths across tea plant plantations ranging from 8 to 43 years old. Soil aggregates became smaller over time and with depth. Soil chemistry changes over time were most prominent near the surface and diminished with depth. C, N, Fe2+, and Mn2+ increased with age, Ca2+ and Mg2+ decreased with age, and P remained stable through time. Soil chemistry changes with soil depth generally occurred in the opposite direction of the changes with age. Although there were anecdotal differences in chemistry among aggregate size classes (e.g., mass fraction of C tended to increase with aggregate size), the changes in aggregate chemistry with depth and over time tended to follow the same patterns as those in the bulk soil. Changes in soil pH were related to Ca:Mg and Fe:Mn ratios, suggesting that soil acidification could be leading to preferential losses of soil micronutrients.

Response: Thank you so much for your time and comments.

The manuscript currently contains six tables and nine figures, which seems a bit overwhelming for most readers to follow. I would strongly suggest that the authors attempt to reduce the amount of raw data presented by identifying the most important aspects of the manuscript. Clarifying the objectives and making the hypotheses more specific would help to provide this focus. In my opinion, the changes in the absolute values of soil nutrients are more relevant than the changes in the ratios of the nutrients (stoichiometry). Therefore, my suggestion would be to move Tables 2-6 into the supplemental and replace them with a single ANOVA table to summarize the stoichiometry findings (e.g., the last 5 rows of Table S1).

Response: Revised. Please see the revised Tables 1-3 and S1-S5, and Figures 2-8.

I believe it is important to provide more details about the site locations and management practices in the materials and methods section. Considering that one of the main aims of this manuscript is to quantify soil nutrient changes through time, it may be necessary to provide some details about the typical annual inputs (e.g., manure, inorganic fertilizers, and litter) and typical outputs (removal of tea for harvest), including approximate annual quantities and nutrient contents. It would also be

helpful to provide more information about the site locations – for example, whether the tea plantations are managed by a single entity or managed independently.

Response: Revised (L 117-131).

The "*Baimao* tea" refers to a major cultivar in such area, and the ages of these tea plantations are distinct. Tea plantations were both experimental trials (Guangxi University) and commercial plantings, and were managed by different owners. In the tea-planting course, tillage method is no tillage and tea-planting density is almost $6 \times 10^4$ plants ha$^{-1}$. Herbicides were not applied and yellow sticky boards were used to prohibit pests, because the color may attract pests and get them stuck on the boards. In addition, all tea plants were subjected to slight pruning in September each year.

An annual fertilizer regime in tea plantations is shown below. Both 0.65 Mg ha$^{-1}$ complex fertilizer (granule, N-P$_2$O$_5$-K$_2$O: 18%-6%-6%) and 12 Mg ha$^{-1}$ swine manure (slurry, N-P$_2$O$_5$-K$_2$O: 0.54%-0.48%-0.36%) were applied yearly in mid-November as the basal fertilizer at the surrounding region vertically below tree crown. Subsequently, the top-dressing, applied to the site treated with replenished basal fertilizer, was replenished 3 times per year. Both 1.2 Mg ha$^{-1}$ complex fertilizer and 0.5 Mg ha$^{-1}$ urea were applied onto soil surface in mid-March, while 0.65 Mg ha$^{-1}$ complex fertilizer and 0.3 Mg ha$^{-1}$ urea were applied in late-June and in early-September.

The authors may want to consider bringing the Figure S1 map into the main text to help with this, as the map shows that the sites are randomly located in space, which helps to mitigate the concern of pseudoreplication.

Response: Revised. Please see the revised Figure 1.

The statistical analyses may require some additional considerations to be sufficiently robust. First, the current two-way ANOVA tests the effects of soil depth and time on the variables (i.e., nutrient concentrations or ratios) within each aggregate size class (e.g., > 2 mm). However, the authors draw many comparisons about differences in those variables among the aggregate size classes (e.g., L221-223), but these differences were not tested statistically. Therefore, my suggestion is to add aggregate size class as an explanatory variable in the ANOVA. Second, for each statistical test, comparisons are being made between all soil depths (4) and times (4), for a total of 16 comparisons. However, according to the statistical methods description, no adjustment is currently being made to compensate for these multiple post-hoc comparisons, and therefore the reported p-values are likely too small. To address this, I suggest using an accepted post-hoc adjustment for multiple comparisons such as Tukey's HSD test. Since this will likely change the significance of some effects, the results and discussion may need to be revised accordingly.

Response: Revised (L 206-211). In this study, SPSS 22.0 was used for statistic analysis. Means were tested by the Tukey's HSD and significance was used at $P < 0.05$. Two-way analysis of variance (ANOVA) was taken for exploring the effects of soil depth, tea plantation age, and their

interactions on the physico-chemical properties of bulk soil. Three-way ANOVA was taken for exploring the effects of soil depth, tea plantation age, aggregate size, and their interactions on the physico-chemical properties of soil aggregates. Moreover, please see the revised Tables (1-3 and S1-S5) and Figures (2-8).

The manuscript requires revisions for grammar.

L1, elsewhere: Suggest changing "stoichiometric characteristics" to "nutrient stoichiometry."

Response: Revised (L 1 and elsewhere).

L1, elsewhere: Suggest changing "tea-planting age" to "tea plantation age."

Response: Revised (L 1 and elsewhere).

L9, elsewhere: "sort of effective way" could be "a tool."

Response: Revised (L 9 and elsewhere).

L14, elsewhere: "at the aggregate scales" could be "within aggregates."

Response: Revised (L 13 and elsewhere).

L22-24: Leaching was not measured in this study, so it seems overreaching to include this in the abstract.

Response: We deleted this inaccurate sentence.

L24-27: The comparison of C and N to other tea plantations is somewhat arbitrary, as soil types may be drastically different among the plantations. I suggest removing these sentences.

Response: We deleted this inaccurate sentence.

L30-32: What is the cause of soil acidification, and how could it be mitigated?

Response: The causes of soil acidification in tea plantation ecosystems were as follows. 1) The losses of soil $Ca^{2+}$ and $Mg^{2+}$, especially the $Ca^{2+}$, could lead to the soil acidification. 2) Tea, as an Al cumulating crop, is able to cumulate Al in leaves. Soil acidification in the tea-planting course was due to the substantial tea litterfall into the soil annually via trimmed branches and leaves. 3) The rhizosphere deposition of massive organic acids (i.e., malate, lemon acid, and oxalate acid) around the tea roots could provoke localized acidification. 4) For increasing the output of tea, tea plantations needed to apply N fertilizers (i.e., urea and $NH_4^+$-N), thus leading to soil acidification by the $NH_4^+$ nitration. In general, reducing the loss of soil $Ca^{2+}$ and increasing the application of organic fertilizer could alleviate soil acidification in tea plantations.

L30-32: Since Mn2+ toxicity was not measured in the study, it does not need to be mentioned in the abstract.

Response: We deleted this inaccurate sentence.

L34: This only provided information about the tea-plantation agroecosystem, not "global terrestrial ecosystems."

Response: Revised (L 28-29).

L56: The term "lower ratio" is ambiguous. Please indicate what is considered high vs. low.

Response: We deleted this inaccurate sentence.

L66-68: Did this study measure total C or organic C? If there were carbonates present at depth, then the total C:N ratio would be much different than the OC:N ratio.

Response: In this study, we measured soil organic C and soil inorganic C was not measured. Because the inorganic C content in most soils is very low, and the turnover period is relatively long. It is mainly a chemical process, which has little relationship with soil fertility, and is often ignored by soil researchers. Moreover, in this study region, the exposed soil horizon occurs early in the Mesozoic, which gradually formed the Ultisols agrotype, and we have not found the carbonates in the deep soil layer.

L85-99: This paragraph seems like it might be more informative at the beginning of the introduction.

Response: Revised (L 34-48).

L107: "assumed" could be "hypothesized."

Response: Revised (L 105).

L129-133: Most readers will be familiar with space-for-time substitution, so it is probably not necessary to describe the concept in great detail here.

Response: We deleted this redundant sentence.

L139-140: "Each of the four tea plantation age groups was replicated in five locations for a total of 20 experimental units"

Response: Revised (L 144-145).

L142: "space self-correlation" could be "spatial autocorrelation."

Response: Revised (L 146).

L146-148: Strictly speaking, this description indicates that the study measured surface litter (a stock), rather than litterfall (a rate). Measuring litterfall would require keeping the newly falling litter separate from the existing surface litter (e.g., litterfall traps) and measuring over a certain period of time (e.g., 1 year).

Response: Revised (L 150).

L152: What was the surface area of each soil sample?

Response: Revised (L 160).

L156, elsewhere: "subjected to filtration" could be "sieved."

Response: Revised (L 162 and elsewhere).

L158-159: Please define all symbols and abbreviations (V and Ø).

Response: Revised (L 165).

L165: Should "vertical" be "horizontal?"

Response: The composite soil specimens were soaked by the aqua destillata for 15 min, and afterwards were oscillated in the VERTICAL direction for 15 min at the 1 s⁻¹ oscillating rate and 5 cm amplitude.

L179, elsewhere: "abstracted" could be "determined."

Response: Revised (L 186 and elsewhere).

L190: Are the "parallel specimens" check standards?

Response: Revised (L 196-198). In this study, 5 standard specimens (GBW-07401), 5 blank specimens, and 80 parallel specimens (accounted for 20% of the total soil specimens) were used to control quality, and the error between parallel specimen and experimental specimen was controlled in 5%.

193-197: Please clarify how the mean weight diameter of aggregates is being used to infer aggregate stability.

Response: Revised (L 201). The mean weight diameter (MWD, mm) was utilized to indicate the stability of soil aggregates. To be specific, if the MWD value is higher, the aggregate stability is stronger.

L202: Please state the alpha value (e.g., $p = 0.05$) used to determine significance.

Response: Revised (L 206-207).

L205-265: P-values should be stated throughout the results section. Any significant interactions should also be noted.

Response: Revised (L 214-278).

L207, elsewhere: "remarkable" or "remarkably" could be "significant" or "significantly" if the implication is that they are statistically significant.

Response: Revised (L 216 and elsewhere).

L224, L249: It is not clear which metric these values were "elevated" above.

Response: Revised (L 232-234 and 257-262).

L231, elsewhere: Differences in nutrient concentrations and ratios among aggregate sizes classes were not statistically tested. Please include statistical tests.

Response: Revised. Please see the revised Tables S1-S5 and Figures 2-8.

L233: It is not clear which source of variation is being discussed with the statement "did not show remarkable variation." For example, does this mean variation by age, by depth, or by aggregate size?

Response: Revised (L 246-247).

L243-244: "mainly distributed" could be "more concentrated."

Response: Revised (L 255-256).

L256-257: "ratios were evenly distributed in" could be "did not vary among."

Response: Revised (L 270).

L262-265: Should Figure 9 be described here?

Response: We have moved this figure into the supplement.

L270-273: What are the potential explanations for higher coarse macroaggregates in the 17-year plantations compared to the other ages? Are the younger plantations managed differently than the older plantations in way that would lead to this outcome? Or was there something specific about that age-group that made them different than the others (e.g., more manure was applied for several years prior to this study)?

Response: Revised (L 286-298). According to the hierarchical concept of soil aggregates, the quality of plant litter returning to the soil determines the distribution of decomposition products of litter in different sizes of aggregates, which ultimately impacts the aggregate composition. In the early (8-17 years) period of tea planting, tea litter displayed greater availability (as indicated by the lower litter C/N ratio), revealing that the decomposition products of litter were easily combined into the coarse macroaggregates, hence fostering the formation of coarse macroaggregates. Reversely, in the middle (17-25 years) and late (25-43 years) periods of tea planting, tea plants naturally encountered aging processes and litter was progressively subjected to humification, which induced the decomposition of coarse macroaggregates into microaggregates. Moreover, the reduced litter amount and covering area after 17 years of tea planting enhanced the rainfall eluviation and artificial interferences (i.e., pruning of tea plants and application of fertilizers), which also caused the destruction of coarse macroaggregates.

L318, elsewhere: "organic matters/OMs" could be "organic matter/OM."

Response: Revised (L 330 and elsewhere).

L322: Please clarify what is meant by "propelled the causal links."

Response: Revised (L 333-334).

L341-346: Changes in soil texture are longer-term processes that would not be expected to change over this time period.

Response: Yes. This study revealed that significant increases in the OC and TN contents were accompanied by no significant variation in the clay content during the process of tea growth, because soil OC and TN contents primarily depend on fertilization, tillage, root exudates, and litter remains, but soil clay content is mainly controlled by its parent material.

L485-503: The conclusions section is nearly identical to the abstract. I would suggest revising the abstract to include a broader opening to provide context for the study (i.e., land use change and tea plantations) and reducing the scope of results given in the abstract.

Response: Revised (L 8-29).

All figure and table captions: Please specify whether the comparisons among different tea

plantations ages (capital letters) are made within each soil depth, and whether the comparisons among each soil depth (lower case letters) are made within each plantation age.

Response: Revised. Please see the revised Figures and Tables.

Table 1: Were other nutrient concentrations measured in the litter (e.g., P, Ca)?

Response: In this study, we just measured the quantity (as indicated by the litter amount, g m-2) and quality (as indicated by the litter C/N ratio) of tea litter, and other nutrients were not measured. However, we believe that the nutrient cycling of litter-soil is well worth further study.

Figures 2, 3, and 4: I suggest converting these line graphs into bar graphs (like figures 5-8) for clarity and consistency.

Response: Revised. Please see the revised Figures 2, 3, and 4.

Figure 9: The regression lines could be colored to match the soil depths to improve interpretability.

Response: Revised. Please see the revised Figure S1.

We believe that we have revised and improved this manuscript to the best of our abilities. In addition, we have made further changes according to the useful and helpful comments you have provided. We sincerely appreciate your time and effort on our behalf, and we truly hope that these corrections will meet with your approval.

Best regards,

Shengqiang Wang

---

## Author Response (AR4)

**First round of modification**

Dear editor Jocelyn Lavallee,

We really appreciate you to give us the chance of revision. Thanks for your comments concerning our manuscript entitled "Dynamics of soil aggregate-related stoichiometric characteristics with tea-planting age and soil depth in the southern Guangxi of China" (SOIL-2021-147). We have made the corrections which we hope will meet with your approval. The revised portions are marked in blue ink in the paper. The main corrections and our responds to the comments are as follows.

Please provide additional detail, and check the vocabulary for clarity (e.g., replace "stochastic" with "random").

Response: Thank you so much for your time and comments. Revised (L 143, 147, and 158).

More detail should be provided for:

* Experimental design: explain the structure and design of the subplots.

Response: As shown in the following figure, for every plot (S = 20 m × 20 m), the 5 litterfall specimens had been acquired from the surface of soil in the 5 randomly chosen subplots (S = 1 m × 1 m), and afterwards were integrated into a composite litterfall specimen. Furthermore, soil sampling was completed in the same sites of the litterfall sampling.

[Figure]

Diagram of quadrat

* Soil sampling: How were samples collected spatially? What is a cutting ring - please describe the size, depth, etc. Provide relevant reference(s) where applicable.

Response: Revised (L 152-154 and 157-160). For every plot, the 5 soil specimens had been acquired by a spade from every soil layer (i.e., 0-10, 10-20, 20-40, and 40-60 cm) in the 5 subplots, and afterwards were integrated into a composite soil specimen. Moreover, extra 5 soil specimens were randomly chosen via cutting rings (V = 100 cm$^{-3}$, Ø = 50.46 mm, and depth = 50 mm) from every soil layer to assess the bulk density, clay (< 0.002 mm), pH, OC, and nutrients of bulk soil.

* Bulk density, texture, pH, and SOC and TN measurements: provide a more detail and relevant references, instrument information, and correction factors where applicable.

Response: Revised (L 172-179). Prior to the analyses of soil physical-chemical properties, soil specimens were subjected to atmospheric drying under indoor temperature condition. According to the cutting ring method (Lu, 2000), soil specimens were oven-dried at 105 °C to the stable weight in order to measure the bulk density. Soil clay was detected by the hydrometer (TM-85, Veichi, China) (Lu, 2000). Soil pH was detected by the glassy electrode (MT-5000, Ehsy, China), with the ratio of soil : water (mass : volume) as 1 : 2.5 (Lu, 2000). Soil OC and TN were identified via the acid dichromate wet oxidation method (Nelson and Sommers, 1996) and the micro-Kjeldahl method (Bremner, 1996), separately. Soil TP was identified via the molybdate blue colorimetry method (Bray and Kurtz, 1945).

* Was inorganic C tested for or measured?

Response: In this study, soil inorganic C was not measured. Because the inorganic C content in most soils, especially the topsoil, is very low, and the turnover period is relatively long. It is mainly a chemical process, which has little relationship with soil fertility, and is often ignored by soil researchers.

**Second round of modification**

Dear editor Jocelyn Lavallee and reviewers,

We really appreciate you to give us the chance of revision. Thanks for your comments concerning our manuscript entitled "Dynamics of soil aggregate-related stoichiometric characteristics with tea-planting age and soil depth in the southern Guangxi of China" (SOIL-2021-147). We have made the corrections which we hope will meet with your approval. The revised portions are marked in blue ink in the paper. The main corrections and our responds to the comments are as follows.

**Reviewer #1:**

The manuscript (Ref. No. soil-2021-147) reported an interesting work on varied stoichiometric characteristics resulted from different tea growing age and soil depth. Such topic fits the scope of the journal very well.

Response: Thank you so much for your time and comments.

However, there are some concerns deserve further clarification before publication. The title needs to be polished due to the unclear expression, e.g., Stoichiometric characteristics of … varied with tea-planting age and soil depth at an aggregate scale in the southern Guangxi of China.

Response: Revised (L 1-2).

Numerous syntaxes and/or grammar problems or misuses existed in the current version, which makes great difficulties in understanding the main points. Native English editing service for the draft was strongly recommended.

Response: We have invited a native English speaker to edit the manuscript in order to improve the logical flow and make the relevant expressions more clear, and also carefully inspected and corrected the details such as word spelling, document information, and English grammar. Please see the revised manuscript.

The research needs or gap for the present study should be clearly indicated and justified as well as the work at the aggregate scale that maybe a potentially important innovative aspect.

Response: Revised (L 79-90). As the basic unites of soil structure, soil aggregates are complex ensembles composed of primary particles as well as organic matter (OM). According to the differences of binding agents, soil aggregates can be classified into microaggregates (< 0.25 mm) and macroaggregates (> 0.25 mm). In general, persistent binding agents (like humified OM and polyvalent metal cation complexes) contribute to the binding of primary particles into microaggregates. Differently, temporary binding agents (like fungal hyphae, plant roots, and polysaccharides) aggregating with microaggregates conduces to the formation of macroaggregates. As shown above, soil aggregates with various sizes exert different abilities in the supply and reserve of soil OC and nutrients. Thus, to improve the comprehension about the structure and function of

soil ecosystems, more efforts should be made to observe the soil stoichiometric characteristics at the aggregate scales.

Some detailed comments for your reference:

P1 Line9, "… a sort of effective way …" should be "… an effective way…".

Response: Revised (L 9).

P1 Line10, "this study was aim to…" changes to "the aim of this study was to…", or "this study was aimed to…", better?

Response: Revised (L 10).

P1 Line 15-16, in various sized aggregates should be in different sizes of aggregates. In different aged tea plantations? Confusing expression. Among different ages of tea gardens or cultivations?

Response: Revised (L 15-16).

P2 Line 27, "an appropriate increase" could be more quantitative or specific?

Response: We deleted this inaccurate sentence.

P2 Line 28, During the process of tea planting, tea growth, better?

Response: Revised (L 24).

P2 Line 31, tea plant should be tea plants or trees. Same as the remaining context.

Response: Revised (L 27 and elsewhere).

**Reviewer #2:**

The manuscript describes a study of soil chemistry within different soil aggregate sizes at various soil depths across tea plant plantations ranging from 8 to 43 years old. Soil aggregates became smaller over time and with depth. Soil chemistry changes over time were most prominent near the surface and diminished with depth. C, N, Fe2+, and Mn2+ increased with age, Ca2+ and Mg2+ decreased with age, and P remained stable through time. Soil chemistry changes with soil depth generally occurred in the opposite direction of the changes with age. Although there were anecdotal differences in chemistry among aggregate size classes (e.g., mass fraction of C tended to increase with aggregate size), the changes in aggregate chemistry with depth and over time tended to follow the same patterns as those in the bulk soil. Changes in soil pH were related to Ca:Mg and Fe:Mn ratios, suggesting that soil acidification could be leading to preferential losses of soil micronutrients.

Response: Thank you so much for your time and comments.

The manuscript currently contains six tables and nine figures, which seems a bit overwhelming for most readers to follow. I would strongly suggest that the authors attempt to reduce the amount of raw data presented by identifying the most important aspects of the manuscript. Clarifying the objectives and making the hypotheses more specific would help to provide this focus. In my

opinion, the changes in the absolute values of soil nutrients are more relevant than the changes in the ratios of the nutrients (stoichiometry). Therefore, my suggestion would be to move Tables 2-6 into the supplemental and replace them with a single ANOVA table to summarize the stoichiometry findings (e.g., the last 5 rows of Table S1).

Response: Revised. Please see the revised Tables 1-3 and S1-S5, and Figures 2-8.

I believe it is important to provide more details about the site locations and management practices in the materials and methods section. Considering that one of the main aims of this manuscript is to quantify soil nutrient changes through time, it may be necessary to provide some details about the typical annual inputs (e.g., manure, inorganic fertilizers, and litter) and typical outputs (removal of tea for harvest), including approximate annual quantities and nutrient contents. It would also be helpful to provide more information about the site locations – for example, whether the tea plantations are managed by a single entity or managed independently.

Response: Revised (L 117-131).

The "*Baimao* tea" refers to a major cultivar in such area, and the ages of these tea plantations are distinct. Tea plantations were both experimental trials (Guangxi University) and commercial plantings, and were managed by different owners. In the tea-planting course, tillage method is no tillage and tea-planting density is almost $6 \times 10^4$ plants ha$^{-1}$. Herbicides were not applied and yellow sticky boards were used to prohibit pests, because the color may attract pests and get them stuck on the boards. In addition, all tea plants were subjected to slight pruning in September each year.

An annual fertilizer regime in tea plantations is shown below. Both 0.65 Mg ha$^{-1}$ complex fertilizer (granule, N-P$_2$O$_5$-K$_2$O: 18%-6%-6%) and 12 Mg ha$^{-1}$ swine manure (slurry, N-P$_2$O$_5$-K$_2$O: 0.54%-0.48%-0.36%) were applied yearly in mid-November as the basal fertilizer at the surrounding region vertically below tree crown. Subsequently, the top-dressing, applied to the site treated with replenished basal fertilizer, was replenished 3 times per year. Both 1.2 Mg ha$^{-1}$ complex fertilizer and 0.5 Mg ha$^{-1}$ urea were applied onto soil surface in mid-March, while 0.65 Mg ha$^{-1}$ complex fertilizer and 0.3 Mg ha$^{-1}$ urea were applied in late-June and in early-September.

The authors may want to consider bringing the Figure S1 map into the main text to help with this, as the map shows that the sites are randomly located in space, which helps to mitigate the concern of pseudoreplication.

Response: Revised. Please see the revised Figure 1.

The statistical analyses may require some additional considerations to be sufficiently robust. First, the current two-way ANOVA tests the effects of soil depth and time on the variables (i.e., nutrient concentrations or ratios) within each aggregate size class (e.g., > 2 mm). However, the authors draw many comparisons about differences in those variables among the aggregate size classes (e.g., L221-223), but these differences were not tested statistically. Therefore, my suggestion is to add

aggregate size class as an explanatory variable in the ANOVA. Second, for each statistical test, comparisons are being made between all soil depths (4) and times (4), for a total of 16 comparisons. However, according to the statistical methods description, no adjustment is currently being made to compensate for these multiple post-hoc comparisons, and therefore the reported p-values are likely too small. To address this, I suggest using an accepted post-hoc adjustment for multiple comparisons such as Tukey's HSD test. Since this will likely change the significance of some effects, the results and discussion may need to be revised accordingly.

Response: Revised (L 206-211). In this study, SPSS 22.0 was used for statistic analysis. Means were tested by the Tukey's HSD and significance was used at $P < 0.05$. Two-way analysis of variance (ANOVA) was taken for exploring the effects of soil depth, tea plantation age, and their interactions on the physico-chemical properties of bulk soil. Three-way ANOVA was taken for exploring the effects of soil depth, tea plantation age, aggregate size, and their interactions on the physico-chemical properties of soil aggregates. Moreover, please see the revised Tables 1-3 and S1-S5, and Figures 2-8.

The manuscript requires revisions for grammar.

L1, elsewhere: Suggest changing "stoichiometric characteristics" to "nutrient stoichiometry."

Response: Revised (L 1 and elsewhere).

L1, elsewhere: Suggest changing "tea-planting age" to "tea plantation age."

Response: Revised (L 1 and elsewhere).

L9, elsewhere: "sort of effective way" could be "a tool."

Response: Revised (L 9 and elsewhere).

L14, elsewhere: "at the aggregate scales" could be "within aggregates."

Response: Revised (L 13 and elsewhere).

L22-24: Leaching was not measured in this study, so it seems overreaching to include this in the abstract.

Response: We deleted this inaccurate sentence.

L24-27: The comparison of C and N to other tea plantations is somewhat arbitrary, as soil types may be drastically different among the plantations. I suggest removing these sentences.

Response: We deleted this inaccurate sentence.

L30-32: What is the cause of soil acidification, and how could it be mitigated?

Response: The causes of soil acidification in tea plantation ecosystems were as follows. 1) The losses of soil $Ca^{2+}$ and $Mg^{2+}$, especially the $Ca^{2+}$, could lead to the soil acidification. 2) Tea, as an Al cumulating crop, is able to cumulate Al in leaves. Soil acidification in the tea-planting course was due to the substantial tea litterfall into the soil annually via trimmed branches and leaves. 3) The rhizosphere deposition of massive organic acids (i.e., malate, lemon acid, and oxalate acid) around

the tea roots could provoke localized acidification. 4) For increasing the output of tea, tea plantations needed to apply N fertilizers (i.e., urea and $NH_4^+$-N), thus leading to soil acidification by the $NH_4^+$ nitration. In general, reducing the loss of soil $Ca^{2+}$ and increasing the application of organic fertilizer could alleviate soil acidification in tea plantations.

L30-32: Since Mn2+ toxicity was not measured in the study, it does not need to be mentioned in the abstract.

Response: We deleted this inaccurate sentence.

L34: This only provided information about the tea-plantation agroecosystem, not "global terrestrial ecosystems."

Response: Revised (L 28-29).

L56: The term "lower ratio" is ambiguous. Please indicate what is considered high vs. low.

Response: We deleted this inaccurate sentence.

L66-68: Did this study measure total C or organic C? If there were carbonates present at depth, then the total C:N ratio would be much different than the OC:N ratio.

Response: In this study, we measured soil organic C and soil inorganic C was not measured. Because the inorganic C content in most soils is very low, and the turnover period is relatively long. It is mainly a chemical process, which has little relationship with soil fertility, and is often ignored by soil researchers. Moreover, in this study region, the exposed soil horizon occurs early in the Mesozoic, which gradually formed the Ultisols agrotype, and we have not found the carbonates in the deep soil layer.

L85-99: This paragraph seems like it might be more informative at the beginning of the introduction.

Response: Revised (L 34-48).

L107: "assumed" could be "hypothesized."

Response: Revised (L 105).

L129-133: Most readers will be familiar with space-for-time substitution, so it is probably not necessary to describe the concept in great detail here.

Response: We deleted this redundant sentence.

L139-140: "Each of the four tea plantation age groups was replicated in five locations for a total of 20 experimental units"

Response: Revised (L 144-145).

L142: "space self-correlation" could be "spatial autocorrelation."

Response: Revised (L 146).

L146-148: Strictly speaking, this description indicates that the study measured surface litter (a stock), rather than litterfall (a rate). Measuring litterfall would require keeping the newly falling

litter separate from the existing surface litter (e.g., litterfall traps) and measuring over a certain period of time (e.g., 1 year).

Response: Revised (L 150).

L152: What was the surface area of each soil sample?

Response: Revised (L 160).

L156, elsewhere: "subjected to filtration" could be "sieved."

Response: Revised (L 162 and elsewhere).

L158-159: Please define all symbols and abbreviations (V and Ø).

Response: Revised (L 165).

L165: Should "vertical" be "horizontal?"

Response: The composite soil specimens were soaked by the aqua destillata for 15 min, and afterwards were oscillated in the VERTICAL direction for 15 min at the 1 $s^{-1}$ oscillating rate and 5 cm amplitude.

L179, elsewhere: "abstracted" could be "determined."

Response: Revised (L 186 and elsewhere).

L190: Are the "parallel specimens" check standards?

Response: Revised (L 196-198). In this study, 5 standard specimens (GBW-07401), 5 blank specimens, and 80 parallel specimens (accounted for 20% of the total soil specimens) were used to control quality, and the error between parallel specimen and experimental specimen was controlled in 5%.

193-197: Please clarify how the mean weight diameter of aggregates is being used to infer aggregate stability.

Response: Revised (L 201). The mean weight diameter (MWD, mm) was utilized to indicate the stability of soil aggregates. To be specific, if the MWD value is higher, the aggregate stability is stronger.

L202: Please state the alpha value (e.g., p = 0.05) used to determine significance.

Response: Revised (L 206-207).

L205-265: P-values should be stated throughout the results section. Any significant interactions should also be noted.

Response: Revised (L 214-278).

L207, elsewhere: "remarkable" or "remarkably" could be "significant" or "significantly" if the implication is that they are statistically significant.

Response: Revised (L 216 and elsewhere).

L224, L249: It is not clear which metric these values were "elevated" above.

Response: Revised (L 232-234 and 257-262).

L231, elsewhere: Differences in nutrient concentrations and ratios among aggregate sizes classes were not statistically tested. Please include statistical tests.

Response: Revised. Please see the revised Tables S1-S5 and Figures 2-8.

L233: It is not clear which source of variation is being discussed with the statement "did not show remarkable variation." For example, does this mean variation by age, by depth, or by aggregate size?

Response: Revised (L 246-247).

L243-244: "mainly distributed" could be "more concentrated."

Response: Revised (L 255-256).

L256-257: "ratios were evenly distributed in" could be "did not vary among."

Response: Revised (L 270).

L262-265: Should Figure 9 be described here?

Response: We have moved this figure into the supplement.

L270-273: What are the potential explanations for higher coarse macroaggregates in the 17-year plantations compared to the other ages? Are the younger plantations managed differently than the older plantations in way that would lead to this outcome? Or was there something specific about that age-group that made them different than the others (e.g., more manure was applied for several years prior to this study)?

Response: Revised (L 286-298). According to the hierarchical concept of soil aggregates, the quality of plant litter returning to the soil determines the distribution of decomposition products of litter in different sizes of aggregates, which ultimately impacts the aggregate composition. In the early (8-17 years) period of tea planting, tea litter displayed greater availability (as indicated by the lower litter C/N ratio), revealing that the decomposition products of litter were easily combined into the coarse macroaggregates, hence fostering the formation of coarse macroaggregates. Reversely, in the middle (17-25 years) and late (25-43 years) periods of tea planting, tea plants naturally encountered aging processes and litter was progressively subjected to humification, which induced the decomposition of coarse macroaggregates into microaggregates. Moreover, the reduced litter amount and covering area after 17 years of tea planting enhanced the rainfall eluviation and artificial interferences (i.e., pruning of tea plants and application of fertilizers), which also caused the destruction of coarse macroaggregates.

L318, elsewhere: "organic matters/OMs" could be "organic matter/OM."

Response: Revised (L 330 and elsewhere).

L322: Please clarify what is meant by "propelled the causal links."

Response: Revised (L 333-334).

L341-346: Changes in soil texture are longer-term processes that would not be expected to change

over this time period.

Response: Yes. This study revealed that significant increases in the OC and TN contents were accompanied by no significant variation in the clay content during the process of tea growth, because soil OC and TN contents primarily depend on fertilization, tillage, root exudates, and litter remains, but soil clay content is mainly controlled by its parent material.

L485-503: The conclusions section is nearly identical to the abstract. I would suggest revising the abstract to include a broader opening to provide context for the study (i.e., land use change and tea plantations) and reducing the scope of results given in the abstract.

Response: Revised (L 8-29).

All figure and table captions: Please specify whether the comparisons among different tea plantations ages (capital letters) are made within each soil depth, and whether the comparisons among each soil depth (lower case letters) are made within each plantation age.

Response: Revised. Please see the revised Figures and Tables.

Table 1: Were other nutrient concentrations measured in the litter (e.g., P, Ca)?

Response: In this study, we just measured the quantity (as indicated by the litter amount, g m-2) and quality (as indicated by the litter C/N ratio) of tea litter, and other nutrients were not measured. However, we believe that the nutrient cycling of litter-soil is well worth further study.

Figures 2, 3, and 4: I suggest converting these line graphs into bar graphs (like figures 5-8) for clarity and consistency.

Response: Revised. Please see the revised Figures 2, 3, and 4.

Figure 9: The regression lines could be colored to match the soil depths to improve interpretability.

Response: Revised. Please see the revised Figure S1.

**Third round of modification**

Dear editor Jocelyn Lavallee and reviewer #2,

We really appreciate you to give us the chance of revision. Many thanks for your comments concerning our manuscript entitled "Soil nutrient stoichiometry varied with tea plantation age and soil depth at an aggregate scale in the southern Guangxi of China" (SOIL-2021-147). We have made the corrections which we hope will meet with your approval. The revised portions are marked in blue ink in the paper. The main corrections and our responds to the comments are as follows.

**Editor:**

Many thanks for your thorough revisions according to the referee reports. The reviewer has some outstanding comments based on your revised version. Please consider these comments and provide a revised version.

Response: Thank you so much for your time and comments. We have invited a native English speaker to edit the manuscript in order to improve the logical flow and make the relevant expressions more clear. Please see the revised manuscript.

[Figure]

**EDITORIAL**

This document certifies that the following manuscript was edited for proper English language by one or more of the highly qualified native English speaking editors at Boyi Translation Co.,Ltd. Tangshan, China.

**Manuscript title:**

Soil nutrient contents and stoichiometry within aggregate size classes varied with tea plantation age and soil depth in the southern Guangxi of China

Date Issued

June 6, 2022

[Figure]

**Reviewer #2:**

The authors have addressed many of the concerns raised during the initial review. I have some additional comments regarding the revisions that the authors may wish to consider.

Response: Thank you so much for your time and comments.

The manuscript readability would be greatly improved with grammatical revisions throughout.

Response: We have invited a native English speaker to edit the manuscript in order to improve the logical flow and make the relevant expressions more clear. Please see the revised manuscript.

Title, elsewhere: this study was performed at a landscape or regional scale, not at an "aggregate scale". A more appropriate term for "aggregate scale" could be "within aggregate size classes."

Response: Revised (L 1 and elsewhere).

Should the title read "Soil nutrient contents and stoichiometry…" to highlight that both contents and stoichiometry were presented in the manuscript?

Response: Revised (L 1).

L28-29, L511-512: The concluding sentence of the abstract and the conclusions section is very general and does not provide the reader with a concrete take-away message. I suggest making these conclusion sentences specific to the study by relating the most significant finding back to the study impetus.

Response: Revised (L 26-28 and L 518-520).

L42-43: By "China is the first nation to plant tea across the globe," do you mean that China is the world's largest producer of tea?

Response: Revised (L 41-42).

L55: It is not clear what is meant by "equilibrium features." Please elaborate upon or rephrase this sentence.

Response: Revised (L 53-55).

L69-76: To interpret the apparent discrepancy in C:N ratios with depth among studies, it is important to consider whether the studies measured total C or organic C. If total C was measured, then C:N ratio might increase with depth due to increasing soil carbonates at depth. However, if organic C was measured, then any carbonates would be eliminated and the C:N ratio would be more likely to decrease with depth unless a buried organic horizon was present.

Response: Revised (L 73-74).

L98-107: The objective and hypothesis are quite general and "observational" in nature. Were there specific questions or hypotheses that were being addressed? For example, "we hypothesized that OC and TN content would increase with tea plantation age …"

Response: Revised (L 104-107).

L139: It is not clear what is meant by "certain underlying mixture effects." Does this mean

"confounding factors?"

Response: Revised (L 139).

L144: A reference to Fig. 1 should be placed here.

Response: Revised (L 145).

L167: The reference to Table 1 and a description of the results within should be made in the results section.

Response: Revised (L 213-221).

L196: It is not clear what is meant by "parallel specimens." Does this mean "analytical replicates" or "analytical duplicates?"

Response: Revised (L 194-195).

L206-207: Please provide a reference for SPSS 22.0.

Response: Revised (L 204).

L218: Please label the Tables and Figures to match the order that they are referenced in the manuscript.

Response: Revised (L 216).

L260-262: The sentence could be revised to clarify what the values are "elevated" above.

Response: Revised (L 268-270).

L365-370: This seems redundant with the previous paragraph. Can this information be combined with the previous paragraph?

Response: Revised (L 374-380).

L410-411: It is not clear what is meant by "triggered by the stratification of humic substance." In addition, the historical concept of humic substances is now mostly considered to be an artifact of the substance extraction procedures. I suggest providing an alternative rational for decreasing C:N ratio with depth, such as older/more processed OM at depth.

Response: Revised (L 419-421).

L506-507: Nutrient management assessments should be performed annually for each site, and therefore broad N fertilization recommendations for the entire region are not appropriate.

Response: Deleted.

Table 2: Please use more conventional symbols to indicate significance level, such as ** for $< p <$ 0.01, * for $p \leq 0.05$, and NS for $p > 0.05$ (i.e., "not significant).

Response: Revised (Table 1).

Figure 1 legend: Please provide more details about the experimental design in the figure legend.

Response: Revised (Figure 1).

Dear editor Jocelyn Lavallee,

We really appreciate you to give us the chance of revision. Many thanks for your comments concerning our manuscript entitled "Soil nutrient contents and stoichiometry within aggregate size classes varied with tea plantation age and soil depth in the southern Guangxi of China" (SOIL-2021-147). We have made the corrections which we hope will meet with your approval. The revised portions are marked in blue ink in the paper. The main corrections and our responds to the comments are as follows.

I appreciate the effort you have put into addressing the reviewer comments and editing the manuscript. I believe this manuscript to be very nearly acceptable for publication, but have some editorial suggestions to improve the readability of this manuscript prior to publication. There are one or two places which require additional details or conceptual changes, which I have also included below.

Response: Thank you so much for your time and comments.

Line 10: remove "the" before "tea plantation"

Response: Removed (L 10) and elsewhere.

Line 17: replace "when" with "as"

Response: Revised (L 17) and elsewhere.

Line 19: remove "the totally"

Response: Removed (L 19).

Line 21: replace "was beneficial for the significant" with "corresponded with"

Response: Revised (L 21).

Line 23: remove "the" before "Ca2+ (as indicated…"

Response: Removed (L 23) and elsewhere.

Line 24: replace ". In the meanwhile, soil acidification" with ", which" (to combine this sentence with the next

Response: Revised (L 24).

Line 34: remove "overmuch"

Response: Removed (L 33).

Line 35: replace "the existing" with "these existing"

Response: Revised (L 34).

Line 38: replace "such" with "this" and "initiated the mode of" with "begun"

Response: Revised (L 36-37).

Line 49: replace "exerts a critical role in recognizing" with "is an invaluable tool for identifying"

Response: Revised (L 47-48).

Line 52: remove "the" before "critical control factors"

Response: Removed (L 50).

Line 55: remove "being"

Response: Removed (L 53).

Line 56: remove "the" before "pivotal"

Response: Removed (L 54).

Line 67: replace "Substantial" with "Several"

Response: Revised (L 65).

Line 68: replace "the increase in" with "increasing"

Response: Revised (L 66) and elsewhere.

Line 69: replace "are discovered" with "have been observed"

Response: Revised (L 67).

Line 70: replace "can be" with "were"

Response: Revised (L 68).

Line 71: replace "the increase in" with "increasing"

Response: Revised (L 69) and elsewhere.

Line 72: replace "is" with "was"

Response: Revised (L 70).

Line 75: replace "displays" with "showed", and "at different soil depths" with "with soil depth", and "as revealed by the" with "in an"

Response: Revised (L 73).

Line 78: replace "unite" with "unit"

Response: Revised (L 76).

Line 82: remove "humified" and replace with a different adjective if necessary which reflects the characteristic of interest. The idea of humification has been discredited. Perhaps something reflecting the size or chemistry of the OM would be more appropriate.

Response: Revised (L 80).

Lines 85-86: remove "Based on the above discussion,"

Response: Removed (L 84).

Line 91: replace "are ended with" with "have shown"

Response: Revised (L 88).

Line 93: replace ". Nevertheless less, some" with "while" (to combine the sentence with the next) and replace "draw the totally" with "observe"

Response: Revised (L 90-91).

Line 94: replace "These demonstrate that" with "Further,"

Response: Revised (L 91).

Line 95: replace "whereas" with "while"

Response: Revised (L 92).

Line 96: replace "rarely" with "less often"

Response: Revised (L 94).

Line 100: replace "deep root" with "deep-rooted", and "reveal" with "understand"

Response: Revised (L 98).

Line 144: replace "totally" with "a total of"

Response: Revised (L 142).

Line 155: "replace "detected" with "measured" and include details of the method and instrument used to make the measurement.

Response: Revised (L 153) and elsewhere.

Line 170: replace "by the aqua distillate" with "in distilled water", and "oscillated" with "shaken"

Response: Revised (L 169).

Line 174: replace "weighted" with "weighed"

Response: Revised (L 173).

Line 195: remove "Besides,"

Response: Removed (L 194).

Line 196: rewrite this sentence as: "The difference between analytical replicates was consistently less than 5%."

Response: Revised (L 194-195).

Line 217: replace "when the" with "as"

Response: Revised (L 216) and elsewhere.

Line 263 and elsewhere: replace "alkali cation" with "alkaline-earth metals". Alkali metals refer to the first group in the periodic table ("The alkali metals are six chemical elements in Group 1, the leftmost column in the periodic table. They are lithium (Li), sodium (Na), potassium (K), rubidium (Rb), cesium (Cs), and francium (Fr)." - https://www.britannica.com/science/alkali-metal)

Response: Thank you so much for your comments. Revised (L 261) and elsewhere.

Line 303: Remove "humification" per Reviewer comments. You can replace with "decomposition".

Response: Revised (L 302).

Table 2: in the explanation of abbreviations, please revise to: "S: soil depth; T: tea plantation age; A: aggregate size. **, *, and NS indicate differences at $p < 0.01$, $p \le 0.05$, and $p > 0.05$ (not significant), respectively.

Response: Revised (Table 1).

We believe that we have revised and improved this manuscript to the best of our abilities. In addition, we have made further changes according to the useful and helpful comments you have provided. We sincerely appreciate your time and effort on our behalf, and we truly hope that these corrections will meet with your approval.

Best regards,
Shengqiang